# Assessing radiation dosimetry for microorganisms in naturally radioactive mineral springs using GATE and Geant4-DNA Monte Carlo simulations

**Sofia Kolovi**[1,2]*, **Giovanna-Rosa Fois**[1], **Sarra Lanouar**[1], **Patrick Chardon**[1,2], **Didier Miallier**[1,2], **Lory-Anne Baker**[2,3,4], **Céline Bailly**[2,5], **Aude Beauger**[2,3], **David G. Biron**[2,3†], **Karine David**[2,5], **Gilles Montavon**[2,5], **Thierry Pilleyre**[1,2], **Benoît Schoefs**[2,6], **Vincent Breton**[1,2‡], **Lydia Maigne**[1,2], with the TIRAMISU Collaboration[¶]

**1** Laboratoire de Physique de Clermont (LPC) - UMR6533, CNRS/IN2P3 Université Clermont Auvergne, Aubière, France, **2** LTSER "Zone Atelier Territoires Uranifères", Clermont-Ferrand, France, **3** Laboratoire Microorganismes: Génome Environnement (LMGE) - UMR6023, CNRS, Université Clermont Auvergne, Clermont–Ferrand, France, **4** Laboratoire de Géographie Physique et Environnementale (GEOLAB) - UMR6042, CNRS, Université Clermont Auvergne, Clermont-Ferrand, France, **5** Laboratoire de Physique Subatomique et des Technologies Associées (SUBATECH) - UMR6457, CNRS/IN2P3/IMT Atlantique/ Université de Nantes, Nantes, France, **6** Metabolism, Molecular Engineering of Microalgae and Applications, Laboratoire de Biologie des Organismes, Stress, Santé Environnement, IUML FR3473, CNRS, Le Mans University, Le Mans, France

† Deceased.
‡ VB is lead author for the TIRAMISU collaboration (vincent.breton@clermont.in2p3.fr).
¶ Membership list of the TIRAMISU collaboration is provided in the Acknowledgments.
* sofia.kolovi@clermont.in2p3.fr

**Data Availability Statement:** The simulation code is available on GitHub (https://github.com/lpc-umr6533/tiramisu_simulation). All relevant

## Abstract

Mineral springs in Massif Central, France can be characterized by higher levels of natural radioactivity in comparison to the background. The biota in these waters is constantly under radiation exposure mainly from the $\alpha$-emitters of the natural decay chains, with $^{226}$Ra in sediments ranging from 21 Bq/g to 43 Bq/g and $^{222}$Rn activity concentrations in water up to 4600 Bq/L. This study couples for the first time micro- and nanodosimetric approaches to radioecology by combining GATE and Geant4-DNA to assess the dose rates and DNA damages to microorganisms living in these naturally radioactive ecosystems. It focuses on unicellular eukaryotic microalgae (diatoms) which display an exceptional abundance of teratological forms in the most radioactive mineral springs in Auvergne. Using spherical geometries for the microorganisms and based on $\gamma$-spectrometric analyses, we evaluate the impact of the external exposure to 1000 Bq/L $^{222}$Rn dissolved in the water and 30 Bq/g $^{226}$Ra in the sediments. Our results show that the external dose rates for diatoms are significant (9.7 μGy/h) and comparable to the threshold (10 μGy/h) for the protection of the ecosystems suggested by the literature. In a first attempt of simulating the radiation induced DNA damage on this species, the rate of DNA Double Strand Breaks per day is estimated to 1.11E-04. Our study confirms the significant mutational pressure from natural radioactivity to which microbial biodiversity has been exposed since Earth origin in hydrothermal springs.

parameters and data are within the paper, allowing the reproducibility of the results.

**Funding:** The current study was performed under S. Kolovi's PhD funding received by the Centre National de la Recherche Scientifique (CNRS) as a "Prime80 CNRS" project with contract No 1083577. The funders had no role in study design, data collection and analysis, decision to publish, or preparation of the manuscript.

**Competing interests:** The authors have declared that no competing interests exist.

## Introduction

Natural radioactivity has been present on Earth since its origin. A growing pattern of evidence suggests that its current levels may affect the mutational load and consequently the genetic composition of plants and animals [1, 2].

Radioactivity also plays an important role in the evolution of the terrestrial microbial biodiversity. At the bottom of mines or beneath the ocean floor, drillings reveal the presence of vast communities of microorganisms in the subsurface of our planet where water radiolysis, following the decay of radionuclides, leads to hydrogen ($H_2$) and oxidants production [3]. This radiolysis could yield enough energy to fuel a large portion of this deep subsurface biome [4]. While ionising radiations have been considered toxic at any level of exposure, experiments at low radiation backgrounds provide a window to explore the contention that responses to radiation dosage are hormetic. Indeed, microbial life is stressed when it is deprived of background levels of radiation [5].

Yet, understanding the role of natural radioactivity in the evolution of microbial biodiversity is methodologically challenging due to its multi-parametric nature [2]. In this perplex context, the assessment of the radioecological risk to the environment due to ionizing radiation has been traditionally addressed through its biota since it is the sensitive component of the ecosystems. Important initiatives, such as ERICA (Environmental Risks from Ionising Contaminants: Assessment and management) tool, provide a number of assessment components following the ICRP (International Commission on Radiological Protection) approach including modelling the transfer of radionuclides through the environment, estimating dose rates to biota from internal and external distributions of radionuclides, and establishing the significance of the dose rates received by organisms [6]. The relevance of ERICA integrated approach has been demonstrated to assess the environmental risks from ionising radiation to macroscopic organisms, but the difficulty of measuring in vivo the dose rates received by biota in the size of a few micrometers in order to assess the potential radiation-induced damages at their DNA (nanoscale) makes essential the use of micro- and nano-dosimetry approaches [7, 8].

A common trend in experimental microdosimetry, as applied in a great variety of fields from aviation and space to nuclear installations and radiation therapy, is the validation of the microdetectors performance by Monte Carlo Simulations (MCS) [9–12]. As it has already been shown in the case of human cells, MCS are needed for micro- and nano-dosimetric assessments due to the stochastic nature of the energy deposition at the cell scale [13].

Aiming to cover the needs of microscale radioecology, we introduce, in this paper, a methodology for modelling the external radiation exposure and its impact on microorganisms living in naturally radioactive aquatic ecosystems using the open-source MCS tools GATE and Geant4-DNA. This work builds upon previous efforts to simulate the impact of the natural radiation background on bacterial systems in the context of very low radiation biological laboratory experiments [14]. We extend it here to eukaryotic microorganisms and to naturally radioactive aquatic ecosystems. An example of such ecosystems is mineral springs, geological formations where microorganisms have been growing in the presence of the radioisotopes of the three natural decays series ($^{238}$U, $^{232}$Th, $^{235}$U), as well as $^{40}$K, since life appeared on Earth [15–18].

The natural decay series of these primordial radionuclides consist of $\alpha$- and $\beta$-emitters, with the former ranging in energy between 4 and 9 MeV. $^{226}$Ra ($t_{1/2}$ = 1600 y) and its gaseous descendant $^{222}$Rn ($t_{1/2}$ = 3.82 d), both $\alpha$-emitters of maximum energy 4.8 MeV and 5.5 MeV respectively [19], are found in high concentrations in the sediments (up to 31 Bq/g $^{226}$Ra) and waters (up to 4600 Bq/L $^{222}$Rn) of the mineral springs in Auvergne [35]. This volcanic region of Massif Central in France is characterized by high uranium content [20] that results in the elevated radium and radon activities measured in local waters [21, 22].

Among other microorganisms living in these peculiar ecosystems, diatoms have received particular attention in recent years. These eukaryotic, photosynthetic, unicellular microalgae are present in marine and freshwater habitats including mineral springs [23, 24] and account for a great part of the carbon dioxide fixation [25, 26]. They vary vastly in shapes and size which can range from a few μm to 2 mm in some cases [27]. What makes them unique to be encountered in the living matter is their frustule, a rigid siliceous cell wall that acts as an external skeleton, remains as fossil after their death [28, 29] allowing studies of their evolution and opens opportunities for their bionanotechnological applications [30–33]. Due to their sensitivity to environmental stresses [34], diatoms are, also, well established as water quality bio-indicators. An exceptional abundance of deformations in the most radioactive springs in Auvergne has been recently revealed, initiating studies of the effects of natural radioactivity on benthic diatom communities in 16 mineral springs of the area [35].

The goal of this paper is to use micro- and nanodosimetric MCS tools to evaluate the dose rates received by the diatoms and the potential induced DNA damage for measured activity concentrations of $^{222}$Rn in the water and $^{226}$Ra in the sediments. GATE is used for the modelling of the radioactive environment and the dose rate assessments, while Geant4-DNA is used for the prediction of Single (SSB) and Double (DSB) DNA Strand Breaks.

GATE (Geant4 Application for Tomographic Emission) is an open-source software based on Geant4 libraries, initially dedicated to medical physics, from imaging to radiotherapy and radiation protection [36–40]. Geant4 (GEometry ANd Tracking) is a simulation toolkit for the passage of particles through matter [41]. Covering a wide range of applications from radiation protection and medical physics to high energy physics, astrophysics and space science, it offers the ability of modelling and simulating from nanoscale up to macroscale [42–46]. Geant4-DNA is dedicated to the simulation of the biological damage in the DNA scale. The set of physics processes used here are adapted to micro- and nano-dosimetry in liquid water allowing the tracking of particles down to eV energies [47–50]. Among others, the assessment of the Single and Double DNA Strand Breaks due to the direct and indirect energy deposition of the ionizing particles is provided through clustering algorithms [51]. In this work, we are engaging the DBSCAN (Density Based Spatial Clustering of Applications with Noise) algorithm [52] to offer a first evaluation of the potential SSBs and DSBs on diatoms due to their chronic exposure to ionizing radiation. For the calculation of DNA damage, DBSCAN takes into account the distribution of deposited energy induced by ionising radiation ($\alpha$-particles in our case) in micrometric geometries and a damage probability function which depends on the total deposited energy.

We first present the measured activity concentrations of the radionuclides of interest in the mineral springs and we gradually build the environmental composition which is essential for the simulation. After detailing the modelling in GATE, we focus on the dose rates received by the microorganisms considering normal benthic and extreme environmental conditions and we study the corresponding effect of the frustule. Then, we couple the GATE results with the Geant4-DNA code to perform the DNA damage simulation and we conclude with the evaluation of the predicted Single (SSB) and Double (DSB) DNA Strand Breaks.

## Materials and methods

The simulation study builds upon the experimental characterization of radionuclides present in water and sediments of five mineral springs in Auvergne, Massif Central, France (Tables 1 and 2). The springs are open for public use and no permit was required to access them and collect samples (Fig 1).

**Table 1. Measured activity concentrations of $^{222}$Rn in water in five mineral springs in Auvergne (Massif central).**

| Spring | Coordinates | Sampling date | $^{222}$Rn (Bq/L) |
|---|---|---|---|
| 1 (Joze) | 45.85057˚N 3.31718˚E | 03/05/2017 | 13.7 ± 0.2 |
| 2 (Joze) | 45.84927˚N 3.31363˚E | 03/05/2017 | 25.3 ± 0.2 |
| 3 (Joze) | 45.85008˚N 3.31826˚E | 03/05/2017 | 421.6 ± 0.6 |
| 4 (Mariol) | 46.02094˚N 3.50589˚E | 13/03/2017 | 147.5 ± 0.9 |
| 5 (Chateldon) | 45.98366˚N 3.53079˚E | 02/03/2017 | 4594.0 ± 2.4 |

**Table 2. Measured mass activities of radionuclides present in sediments in five mineral springs in Auvergne (Massif central).**

| Spring | Sampling Date | $^{226}$Ra (Bq/g) | $^{238}$U (Bq/g) | $^{228}$Ra (Bq/g) | $^{228}$Th (Bq/g) |
|---|---|---|---|---|---|
| 1 (Joze) | 03/05/2017 | 30.8 ± 0.6 | 3.9 ± 0.1 | 9.6 ± 0.4 | 5.5 ± 0.3 |
| 2 (Joze) | 03/05/2017 | 42.5 ± 0.9 | 4.5 ± 0.1 | 13.9 ± 0.6 | 6.7 ± 0.4 |
| 3 (Joze) | 03/05/2017 | 21.4 ± 0.4 | 2.3 ± 0.1 | 1.6 ± 0.1 | 1.1 ± 0.1 |
| 4 (Mariol) | 23/03/2017 | 31.9 ± 0.6 | 3.7 ± 0.1 | 14.7 ± 0.6 | 3.4 ± 0.1 |
| 5 (Chateldon) | 01/03/2017 | 31.4 ± 0.6 | 5.4 ± 0.2 | 1.5 ± 0.1 | 0.4 ± 0.1 |

In two of these springs—Mariol and Chateldon—the deformation rate of different diatom species was documented as a function of the water $^{222}$Rn content [35]. In the following subsections, we first describe how the aquatic environment and the diatoms have been characterized, to then, elaborate the multi-scale simulations using GATE and Geant4-DNA softwares.

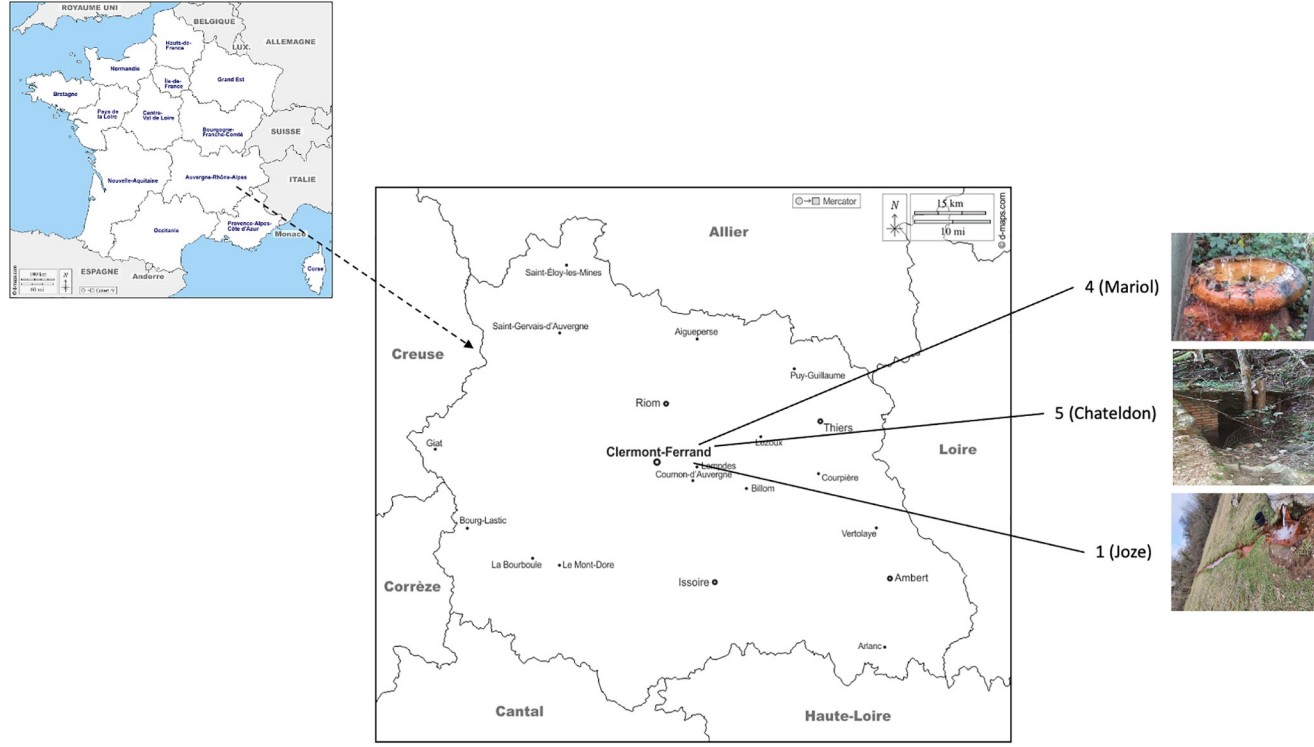

**Fig 1. Location of springs 1 (Joze), 4 (Mariol) and 5 (Chateldon).** Maps are reprinted from [53] under a CC BY license, with permission from Daniel Dalet, original copyright 2023.

## Radiological characterization of the mineral springs

The activity concentrations of $^{222}$Rn in water and $^{226}$Ra in the sediments from five mineral springs were measured by $\gamma$-spectrometry using a High-Purity Germanium (HPGe) well-type detector (GCW3523,Canberra Inc., Toledo, USA) of 35% relative efficiency. Water was collected according to ISO 5667-1 and ISO 5667-3 standards in Marinelli style gas-analysis containers (NUVIA Instruments GMBH) designed for $\gamma$–spectroscopic analysis. The beakers were sealed to avoid any $^{222}$Rn leakage. $^{222}$Rn activity concentrations were measured within the first 3 hours after the sampling using the 352 keV $\gamma$-ray of $^{214}$Pb ($t_{1/2}$ = 27.06 min). The sediment samples were dried in a laboratory fume hood until constant mass, sieved to remove parts greater than 2 mm and finally sealed, according to ISO 18589-2. By weighting the sediment sample before and after the drying process, we estimated the environmental medium to be a mixture composed by 90% water and 10% dry sediments. The measurements of the $^{226}$Ra mass activity were performed at least 4 weeks after the sealing to allow the secular equilibrium between $^{226}$Ra ($t_{1/2}$ = 1600 y) and the immediate $^{222}$Rn ($t_{1/2}$ = 3.82 d) daughters ($^{214}$Bi—$t_{1/2}$ = 19.9 min, $^{214}$Pb—$t_{1/2}$ = 27.06 min) to be reached. The 609.3 keV $\gamma$-peak of $^{214}$Bi and the 295, 352 keV $\gamma$-peaks of $^{214}$Pb were, then, used to determine the $^{226}$Ra activity in the sediments.

The range of the measured activity concentrations for $^{222}$Rn in water and $^{226}$Ra in sediments are presented in Tables 1 and 2, together with the activity concentrations of $^{238}$U and two of the $^{232}$Th decay chain daughters; $^{228}$Ra and $^{228}$Th. A common feature of several $CO_2$-rich geothermal systems is a high high radium content acquired during the ascent of water from the deep reservoir through the crystalline bedrock. High radium solubility in the mineral waters can be related to their major ion composition [21]. As a consequence, the sediments and deposits (travertine) precipitated from the waters display a disequilibrium in the uranium and thorium decay chains. Millan et al. [35] established a first correlation between $^{222}$Rn activity measured in water of 17 mineral springs in Auvergne (including the ones considered in this study) and teratological forms of diatoms. They remark that significant rates of deformations are observed for activity concentrations higher than 1000 Bq/L for $^{222}$Rn in water and 30 Bq/g for $^{226}$Ra in sediments; therefore, those two values were kept as reference for the simulation studies.

An X-Ray fluorescence analysis using a Niton XL5 analyzer (Thermo Fisher Scientific Inc., Waltham, USA) was performed for the determination of the mass fraction of the mean elementary composition of the sampled sediments (Table 3). Carbon, oxygen, calcium and silicon account for approximately 95% of the mass fraction with the rest consisting of heavier metals such as iron, aluminum, potassium, magnesium, strontium and titanium, and a smaller contribution of chlorine, sulfur and phosphorus.

## Characterization of the diatoms communities

Diatoms belong to a rich ecosystem of microorganisms including bacteria and viruses that thrive in mineral springs. The benthic species inhabit rocks or sediments of the springs floor

**Table 3. Dry sediments composition using X-Ray fluorescence analysis.**

| Element | % | Element | % |
|---|---|---|---|
| C & O | 60.36 | K | 0.70 |
| Ca | 24.50 | Mg | 0.53 |
| Si | 10.00 | Sr | 0.50 |
| Fe | 1.60 | Ti | 0.30 |
| Cl | 1.40 | S | 0.30 |
| Al | 1.00 | P | 0.08 |

but they can also be found in the water column. Small samples of epipelic and epilithic raw material were prepared for light microscopy (LM) observations, morphometric measurements and evaluation of the relative abundance of diatom species. Diatoms were imaged using an ultra high-resolution analytical field emission scanning electron microscope (SEM) Hitachi SU-70 (Hitachi High-Technologies Corporation, Japan). SEM images revealed various abnormal forms of diatoms and information concerning their dimensions were used to build a model for the simulation purpose. Ellipsoidal dimensions of individuals vary between 5 to 50 μm for the major axis and 4 to 7 μm for the minor axis. The study by Beauger et al. [23] provides details about 18 dominant diatom species with relative abundance higher than 1% in 17 springs located near Clermont-Ferrand close to Allier river. *Planothidium frequentissimum*, *Navicula sanctamargaritae* and *Crenotia thermalis* are three of the species with an abundance exceeding 60%. The same species are observed at spring 5 (Chateldon) where deformation rates above 25%, denoted by abnormally shaped frustules, have been observed on *Planothidium frequentissimum* [35]. Diatom deformations, or else stated teratological formations, are abnormalities on their morphology, mainly on the valve shape and structural characteristics which result in deformed frustules.

Genomics information for diatoms living in Auvergne mineral springs is currently missing. Indeed, only ten out of 200 000 diatom species have been completely sequenced up to day, revealing a range in DNA size between 27 and 162 Mbp preserved in 1—2 μm diameter nuclei [54, 55].

## Dosimetry simulation using GATE

Absorbed dose rates to a diatom have been computed with GATE v9.1 using Geant4 v.11.0.0 libraries. Being the most abundant constituent of cells, water is considered as a surrogate to the biological medium [56, 57]. In our simulation, the microorganism and its nucleus were modelled as water spheres of 10 $\mu$m ($r_M$) and 0.5 $\mu$m radius respectively. For the modelling of the diatom, a $SiO_2$ (Silicate) shell of 2 $\mu$m thickness (F) was added around the microorganism, representing their frustule (rigid exoskeleton).

The environment surrounding the microorganism was modelled as a sphere with a radius ($R_{env}$) calculated according to the following formula:

$$R_{env} = (R_{max} + r_M + F) \cdot 1.03 \tag{1}$$

where $R_{max}$ represents the range in water of the most energetic $\alpha$-particles in the simulation, ($R_{max} = 43.44$ $\mu$m for $^{222}$Rn [58]) and the multiplication factor 1.03 offers an extra 3% space margin. The composition of the environment can be either water, dry sediments (Table 3) or a mixture of water and dry sediments, denoted as "benthic mixture"(BM). A percentage porosity (P), defined in Eq 2, is used to characterize each simulated environment:

$$P(\%) = \frac{V_W}{V_{tot}} \cdot 100 \tag{2}$$

where $V_W$ is the volume of water and $V_{tot}$ is the total volume of the mixture. In this work, we focus on three porosity values to define the environment: "0%", "90%" and "100%". The "0%" corresponds to an environment made only with dry sediments, the "100%" with only water, while "90%" porosity refers to the observed conditions at the bottom of the water column where benthic diatoms develop. Microorganisms living in the water column are typically exposed to doses corresponding to 100% porosity while the benthic ones, living on the floor of the springs or on rocks, are typically exposed to the 90% porosity scenario. It should be clear that the dry sediment scenario corresponding to 0% porosity does not reflect a relevant

**Table 4. Simulated $\alpha$-particle energies and intensities from $^{222}$Rn [60] and $^{226}$Ra [61].**

| Radionuclide | $E_\alpha$ (MeV) | Intensity (%) |
|---|---|---|
| $^{222}$Rn | 4.826 | 5.0E-04 |
| | 4.986 | 0.078 |
| | 5.490 | 99.92 |
| $^{226}$Ra | 4.160 | 2.7E-04 |
| | 4.191 | 1.0E-03 |
| | 4.340 | 6.5E-03 |
| | 4.601 | 6.16 |
| | 4.784 | 93.84 |

environment for diatoms inhabiting mineral springs and should be considered as an upper limit in this context. However, diatoms have been observed living outside water [59] and some springs dry up during summer seasons.

This study focuses on the dose rates to microorganisms coming from radioelements that have been measured experimentally to have the highest activity concentrations in the sediments and waters of Auvergne mineral springs. As a result, we simulated only the $\alpha$-particles emitted directly by $^{222}$Rn and $^{226}$Ra (see Table 4). These radioelements also decay and their daughters, especially the $\alpha$-emitters, contribute an additional radiation dose to the microorganisms. In a first step, this contribution was not computed using Monte-Carlo simulations because the chemical behaviour of these radionuclides and therefore their location in the vicinity of the diatoms is not known. We verified by preliminary simulations that $\beta$-emitters in the $^{238}$U decay chain, mainly $^{214}$Bi and $^{210}$Pb, could be neglected. Their contribution to the dose is 0.02% in comparison to the $\alpha$-emitters and they were, consequently, not taken into consideration in the simulation. For each radionuclide, the $\alpha$-particles were emitted isotropically ($4\pi$ solid angle) from the spherical volume surrounding the microorganism where their emission point was randomly distributed. Separate simulations were performed for $^{222}$Rn and $^{226}$Ra. For pure dry sediments (0% porosity), $^{226}$Ra is the only source of radioactivity, while for pure water (100% porosity) only $^{222}$Rn is considered. Both of the radionuclides are present in the "benthic mixture" (90% porosity).

In all simulations we used the Geant4 electromagnetic physics list option 4. The production cuts applied to secondary electrons and gammas, which are produced due to the interactions of the $\alpha$-particles with the matter, were investigated in preliminary simulations, and chosen to be 2 orders of magnitude less than the size of radius of the simulated volumes: 0.1 $\mu$m in the environment, 0.01 $\mu$m in the diatom (microorganism plus frustule), and 0.001 $\mu$m in the nucleus, corresponding thus to the lowest cut-off energy available in GATE (250 eV). The setup of the simulation is summarized in Table 5. G4_WATER was used as material in all cases [62].

**Table 5. Summary of GATE simulation characteristics.**

| | environment (frustule*) | microorganism | nucleus | * frustule |
|---|---|---|---|---|
| **Shape** | Sphere | Sphere | Sphere | Shell |
| **Size** | R = 55 $\mu$m (57.1 $\mu$m) | R = 10 $\mu$m | R = 0.5 $\mu$m | Width = 2 $\mu$m |
| **Material** | G4_WATER / dry sediments / mixture | G4_WATER | G4_WATER | Silicate |
| **Density (g/cm$^3$)** | 1.00 / 1.20 / 1.02 | 1.00 | 1.00 | 2.40 |
| **Cuts ($\mu$m)** | 0.1 | 0.01 | 0.001 | 0.01 |
| **Source** | $^{222}$Rn / $^{226}$Ra | - | - | - |
| **Physics List** | electromagnetic standard option 4 | | | |

First, information concerning particles (energy, position, direction and particle type) entering the microorganism were recorded in a phase space (PhSp) file attached to the external boundary of the microorganism. Then, the energy deposited to the microorganism for each radionuclide was recorded, the contributions of the dominant physical processes were identified, and the total energy depositions were calculated. Absorbed dose rates to the microorganism in µGy/h were calculated for every porosity level while the silicate frustule was taken into account for the simulation of the "benthic mixture" (90% porosity). We ran separate simulations with 1E+08 primary $\alpha$-particles for each radionuclide and repeated them 10 times to evaluate the statistical fluctuations (kept below 1%). We, then, scaled the dose rates obtained to the primaries corresponding to the reference activity values. Information concerning $\alpha$-particles entering the nucleus were recorded in a PhSp file attached to the external boundary of the nucleus in order to be used as source description for Geant4-DNA simulations.

## Simulation of DNA damage using Geant4-DNA

The "G4EmDNAPhysics_option4" in Geant4 version 11.0.0 was used to simulate track structure of particles inside the microorganism nucleus from the PhSp file produced at the previous step. This physics list is considered the most accurate for the simulation of low energy electrons transport in water [63]. The production cuts for electrons and photons were set to 1 nm. A summary of the Geant4-DNA simulation setup is presented in Table 6.

Using the $\alpha$-particles generated from the PhSp file as source, we collected the distribution of deposited energies using the simplified model suggested by D.E. Charlton [64]. We simulated 30 000 water cylinders representing nucleosomes (approximately 147 base pairs each) of 10 nm diameter and 5 nm height made of water. All G4-DNA models are validated and available for water (G4_WATER as defined in NIST [62]) which is used as a surrogate to the biological medium. The cylinders were generated in random positions in the spherical nucleus of the microorganism. Energy deposits were collected in the 30 000 randomly distributed cylindrical targets and the probability distributions of the specific energies (dP/dz) over all the nucleosomes were calculated.

Specific energies usually result from several energy transfers in a given nanometric target and therefore from several physical processes. We can, however, assign a physical process to each specific energy by considering the dominant process which leads to the largest contribution of energy deposition. This allows us to study the total specific energy spectra and the different contributions from dominant processes.

The specific energy (SE) rates (µGy/h) per nucleosome were obtained by averaging the specific energies over the total number of the cylindrical targets.

To assess radiation induced SSBs and DSBs, the DBSCAN clustering algorithm was utilized [52]. It is based on the assumption that the nucleus is occupied by uniformly distributed DNA molecules, which is actually indicative of diatoms presenting an active metabolic activity [65],

**Table 6. Summary of Geant4-DNA simulation characteristics.**

|  | nucleus | nucleosomes |
|---|---|---|
| Dimensions | radius: 500 nm | diameter: 10 nm height: 5 nm |
| Material | | G4_WATER |
| Density (g/cm$^3$) | | 1.00 |
| Cuts (nm) | | 1.00 |
| Source | | $\alpha$-particles from PhaseSpace |
| Physics Lists | | G4EmDNAPhysics_option4 |

thus making it possible to predict the potential DNA damage without the use of sophisticated DNA geometry. Such geometry is not available for the diatoms inhabiting the mineral springs because their genome has not been sequenced yet. Because diatoms living in mineral springs are smaller than the marine diatoms sequenced, we made the assumption of a 1 μm diameter nucleus enclosing 27 Mbp of genetic material.

The formation of SSBs in this algorithm is a function of the energy deposited following a probability distribution function. For deposited energies (edep) below 5 eV the damage probability is considered zero while it increases linearly up to 37.5 eV. For edep $\geq$ 37.5 eV the algorithm considers that all the events can cause SSBs. The minimum number of SSBs to form a DSB is set to 2 within a radius of 3.3 nm, representing roughly the distance between 10 DNA base pairs. The indirect DNA damage due to the radicals formation after water radiolysis is taken into consideration in the free parameter "SPointProb". It describes the probability that an interaction point is located in a sensitive area, composed by the DNA helix and a virtual aura, where both direct and indirect DNA damage can occur. Strong correlations between the genome size and phenotypic characteristics, such as nuclear and cell volume, are abundantly documented in the literature for eukaryotes [66]. As a consequence, we tested different values of the free parameter "SPointProb" in order to validate the algorithm against simulation and experimental data from the literature on human cells. For the source description, we considered the PhaseSpace and the physics list described in the previous section. We ran the simulation multiple times, achieving a relative uncertainty below 0.1%.

## Results

### Absorbed dose to microorganisms

By recovering the number of particles entering the different volumes, we observed that only 2% of the primaries emitted in the 55 μm radius environment reached the microorganism, while the presence of the frustule resulted in an extra 20% decrease in the number of particles entering the diatom. In Table 7 we summarize the recorded kinetic energies of the $\alpha$-particles (primaries) reaching the microorganism for the different simulated environments. The most energetic primaries have a mean energy of 3.3 MeV coming from $^{222}$Rn dropping by 12% when considering the frustule. A similar trend is observed for $^{226}$Ra with a mean energy of 2.5 MeV.

Fig 2 shows the distribution of deposited energies to the diatom in the benthic mixture (90% porosity). The main physical processes involved in the energy depositions are ion and electron ionisations. Ion ionisation refers to the ionisations caused directly from the $\alpha$-particle while the electron ionisation refers to the ionisations caused by sufficiently energetic secondary electrons ($\delta$-rays). We show that the predominant process is ion ionisation while the mean energy deposition due to electrons ionisation is merely 15 keV.

**Table 7. Kinetic energy of $\alpha$-particles reaching the microorganism for different environments (when considering frustule, values are provided in the parentheses).**

| Porosity (%) | Environment | Energy (MeV) | |
| --- | --- | --- | --- |
| | | mean | maximum |
| 0 | Dry sediments—$^{226}$Ra | 2.8 | 4.8 |
| 90 | Benthic Mixture—$^{226}$Ra (frustule) | 2.8 (2.5) | 4.8 (4.5) |
| 90 | Benthic Mixture—$^{222}$Rn (frustule) | 3.3 (2.9) | 5.5 (5.2) |
| 100 | Water column—$^{222}$Rn | 3.3 | 5.5 |

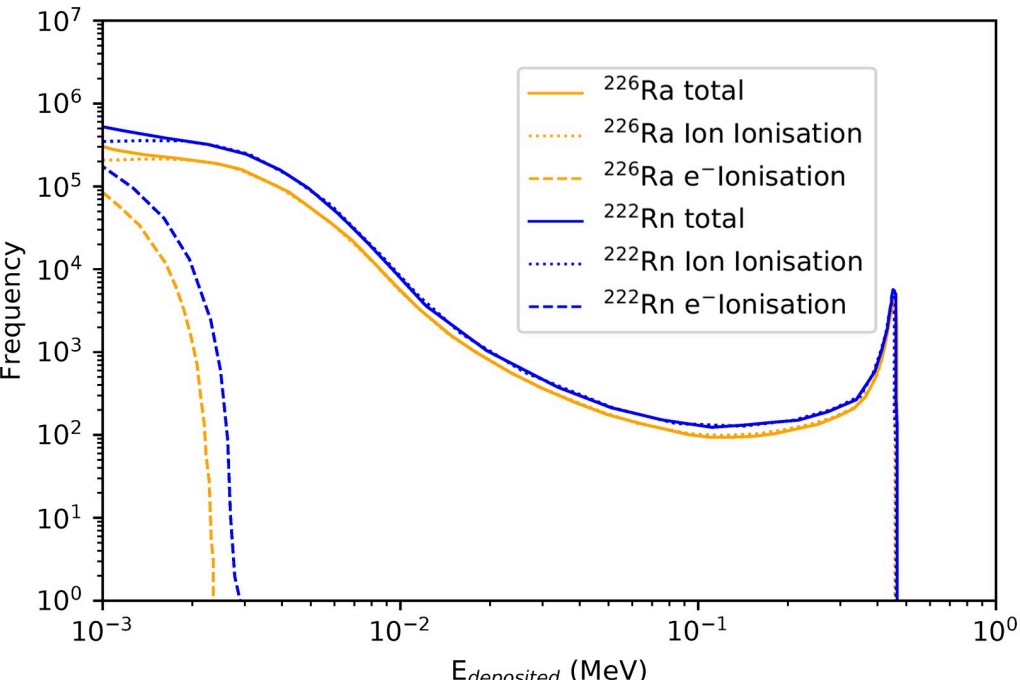

**Fig 2. Deposited energy distributions for $^{226}$Ra (orange) and $^{222}$Rn (blue) in the benthic mixture for diatoms.** Solid lines: total deposited energy, Dotted lines: ion ionisation, Dashed lines: electron ionisation.

In Table 8, the absolute doses for 1E+08 primaries are slightly higher for $^{222}$Rn in comparison to $^{226}$Ra. We remark that the frustule is responsible for an average 20% absolute dose decrease. The dose rates to the microorganisms, normalised to the reference realistic radiological conditions, are summarized in the same table. The simulation showed that a microorganism surrounded only by dry sediments (containing only $^{226}$Ra) would be exposed to 92.4 μGy/h whereas in the scenario of a sole aquatic environment containing only $^{222}$Rn the respective value gets reduced to 2.8 μGy/h.

In Fig 3 we present the dose rates to microorganisms without (10.8 μGy/h) and with frustule (9.7 μGy/h) for 1000 Bq/L $^{222}$Rn in water and 30 Bq/g $^{226}$Ra in the dry sediments. We can observe the diminution of the total dose rate due to the frustule which is equal to 10%. We highlight also that dose rates from $^{222}$Rn accounts only for one third of the total ones.

**Table 8. Absolute doses and normalised dose rates to microorganisms in all environments (when considering frustule, values are provided in the parentheses).**

| Porosity (%) | Environment | Absolute dose (Gy) | Dose Rate (μGy/h) |
|---|---|---|---|
| 0 | Dry sediments—$^{226}$Ra | 10.3E+04 | 92.4 |
| 90 | Benthic Mixture—$^{226}$Ra (frustule) | 9.3E+04 (7.3E+04) | 8.3 (7.4) |
| 90 | Benthic Mixture—$^{222}$Rn (frustule) | 11.1E+04 (9.2E+04) | 2.5 (2.3) |
| 100 | Water column—$^{222}$Rn | 11.2E+04 | 2.8 |

Absolute doses correspond to 1E+08 primaries. Dose rates are normalised to 1000 Bq/L $^{222}$Rn in water and 30 Bq/g $^{226}$Ra in the dry sediments.

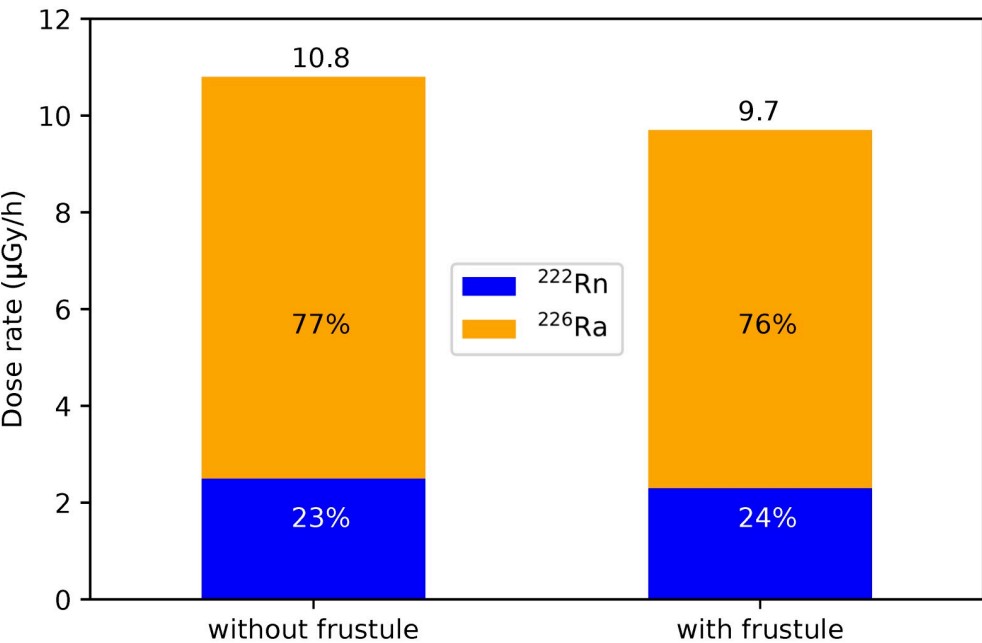

**Fig 3. Dose rates to diatom in the benthic mixture.** Blue bar: [222]Rn contribution, Orange bar: [226]Ra contribution.

## DNA damage in microorganisms

From the 1E+08 primaries emitted in the environment, only very few alphas (0.003%) reached the nucleus. In Table 9 we present the mean and maximum kinetic energies of the $\alpha$-particles entering the nucleus, as well as the mean energy they deposited to the nucleus, for the different simulated environments. The $\alpha$-particles of [226]Ra reach the nucleus with a 25% reduced mean energy compared to their initial emission in the environment, while with the frustule the loss is about 32%. A similar trend is observed for [222]Rn with 18% loss. We also notice that the frustule has a very limited impact on the deposited energy.

Fig 4 presents the total specific energy probability distribution, while Figs 5 and 6 present the distributions associated to the main physical processes in the benthic mixture (90% porosity considering the frustule). The main contributions come from He ions and electrons. We no further distinguish [222]Rn and [226]Ra in the benthic mixture (90% porosity) for the nanodosimetric assessment.

The DSB/Gy/Mbp for four different SPointProb values of the free parameter in the DBSCAN algorithm for monoenergetic $\alpha$-particles are plotted against the available literature [67] in Fig 7. As it is shown, our predicted values are in general agreement with the experimental

**Table 9. Kinetic and deposited energy of $\alpha$-particles at the nucleus for different environments (when considering frustule, values are provided in the parentheses).**

| Porosity (%) | Environment | Kinetic energy (MeV) | | Deposited energy (MeV) |
|---|---|---|---|---|
| | | mean | maximum | mean |
| 0 | Dry sediments—[226]Ra | 2.1 | 3.9 | 1.4E-03 |
| 90 | Benthic Mixture—[226]Ra (frustule) | 2.1 (1.9) | 3.9 (3.5) | 1.4E-03 (1.4E-03) |
| 90 | Benthic Mixture—[222]Rn (frustule) | 2.7 (2.4) | 4.7 (4.4) | 1.3E-03 (1.3E-03) |
| 100 | Water column—[222]Rn | 2.7 | 4.7 | 1.3E-03 |

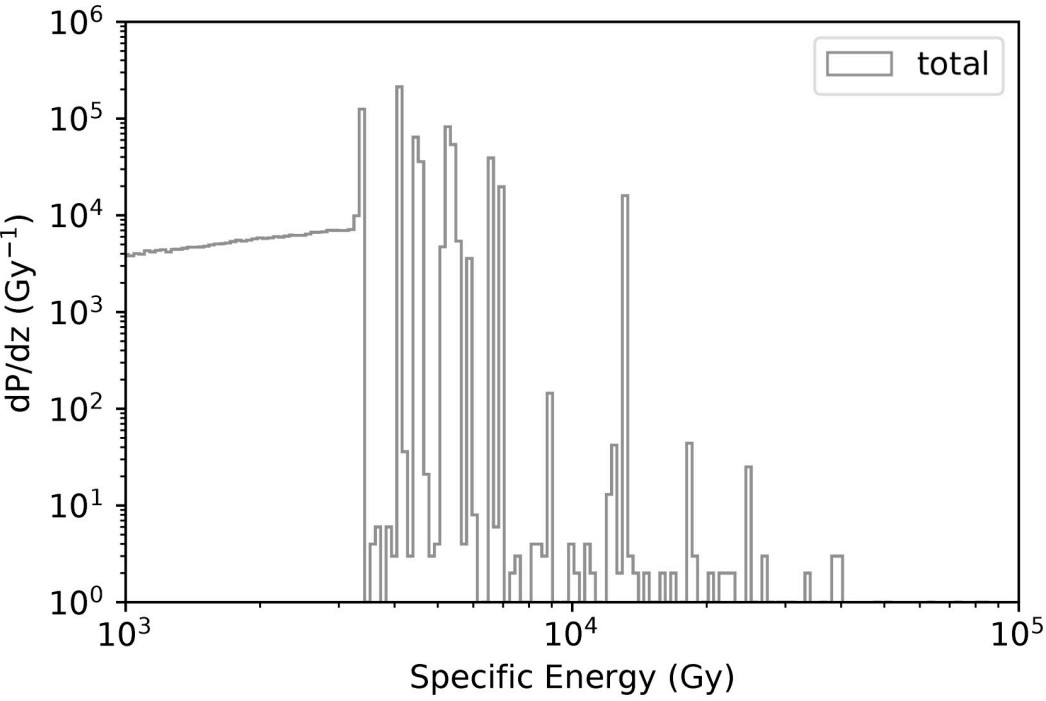

**Fig 4. Total specific energy probability distribution (Gy$^{-1}$) for nucleosomes (90% porosity).** Grey: All processes.

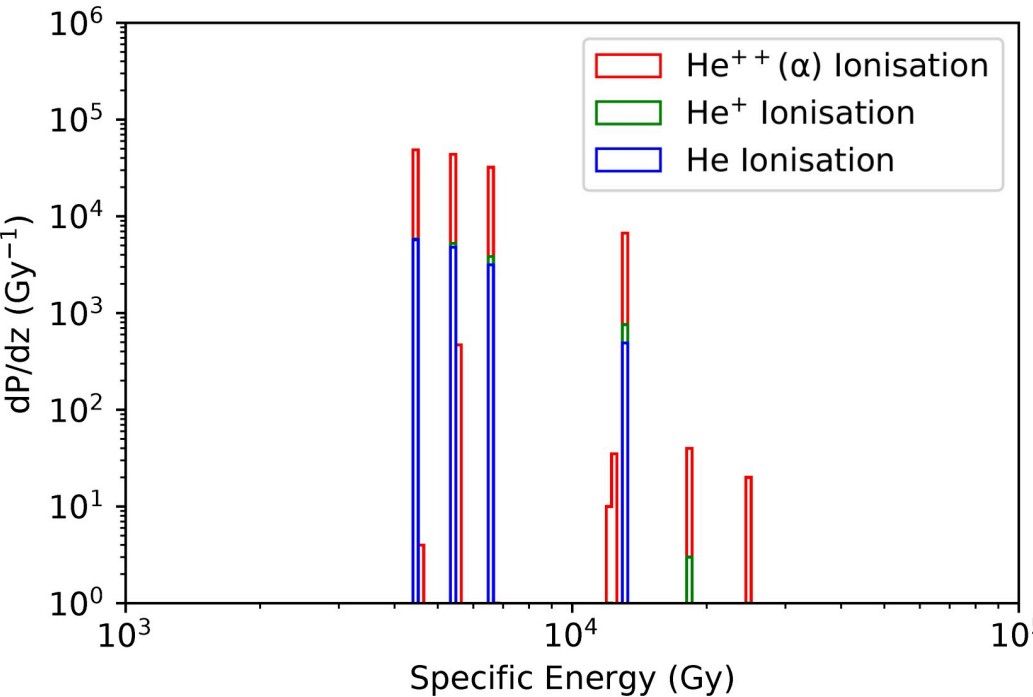

**Fig 5. He ions specific energy probability distributions (Gy$^{-1}$) for nucleosomes (90% porosity).** Red: $\alpha$-particles (He$^{++}$) ionisation, Green: He$^{+}$ ionisation, Blue: He ionisation.

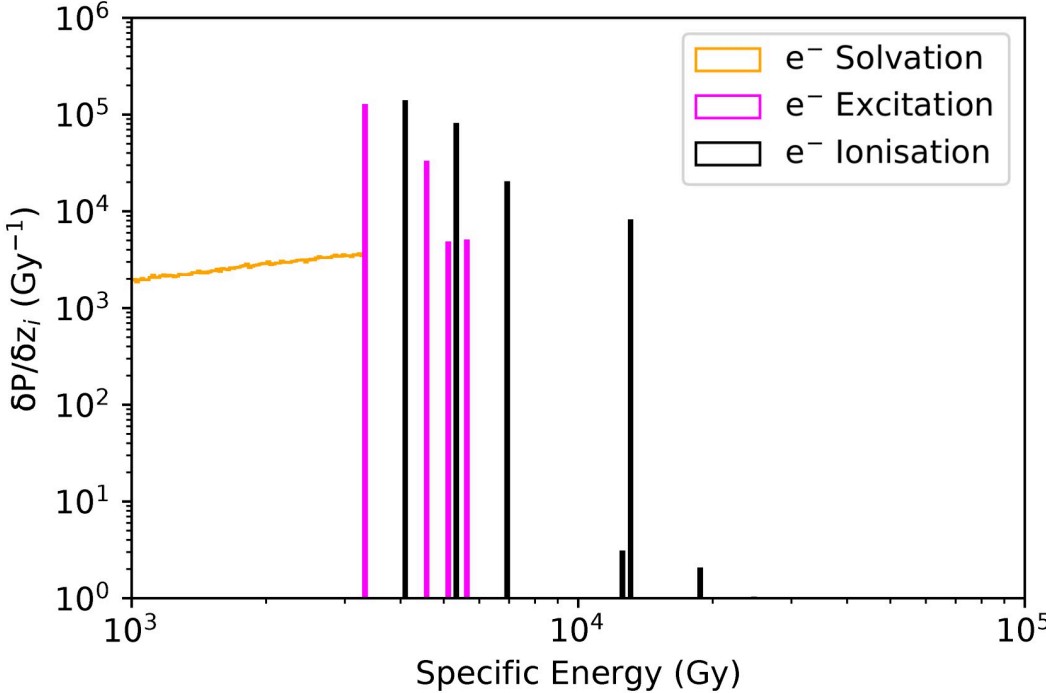

**Fig 6. Electrons specific energy probability distributions (Gy⁻¹) for nucleosomes (90% porosity).** Orange: electrons solvation, Purple: electrons excitation, Black: electrons ionisation.

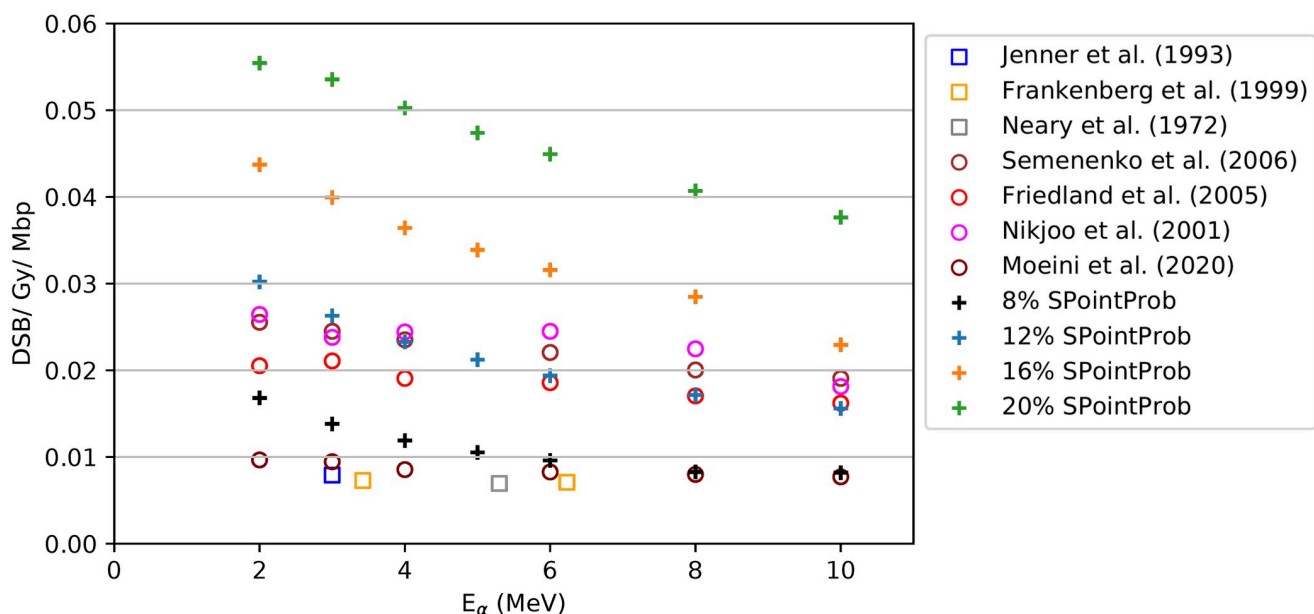

**Fig 7. Number of DSB per Gray per Mbp as a function of the energy of $\alpha$-particles.** (+): different values of the SPointProb parameter in our work, (○): simulations using other codes found in the literature [67–70], (□): experiments found in the literature [71–73].

**Table 10. Mean specific energy (SE) rates per nucleosome (μGy/h) and number of SSBs, DSBs (when considering frustule, values are provided in the parentheses).**

| Porosity (%) | 0 | 90 | 100 |
|---|---|---|---|
| Environment | Dry sediments—$^{226}$Ra | Benthic Mixture—$^{226}$Ra & $^{222}$Rn (frustule) | Water column—$^{222}$Rn |
| SE rate (μGy/h) | 71.70 | 8.31 (7.36) | 2.12 |
| SSB/Gy/Mbp | 0.07 | 0.16 (0.15) | 0.08 |
| DSB/Gy/Mbp | 0.02 | 0.03 (0.03) | 0.02 |
| SSB/day | 4.50E-03 | 5.40E-04 (4.70E-04) | 1.48E-04 |
| DSB/day | 1.06E-03 | 1.21-04 (1.11E-04) | 2.99E-05 |

The SSB/day and DSB/day are normalised for 1-day exposure to 30 Bq/g $^{226}$Ra, 1000 Bq/L $^{222}$Rn and 27 Mbp DNA length

and other simulation results using different MC codes. The overprediction of DSBs when using the suggested values of 16% [52] and 20% [48] smooths when lowering the value. 12% offers a good agreement with H. Moeini et al. [67] but we finally chose SpointProb equal to 8% due to the better agreement with the experimental results, too.

In Table 10, the mean specific energy (SE) rates to a nucleosome and the SSBs and DSBs obtained per Gray and per Mbp using the DBSCAN clustering algorithm (8% SPointProb) for the different simulated environments are listed. In the same table, we present the SSBs and DSBs per day normalised to 27 Mbp and, to 1 day-exposure to 30 Bq/g $^{226}$Ra in dry sediments and 1000 Bq/L $^{222}$Rn in water.

We observe that the frustule decreases the mean specific energy rates by 11%. We also highlight that the nucleosomes of microorganisms living in dry sediments containing exclusively 30 Bq/g of $^{226}$Ra are exposed to 34 times higher specific energies rates than the microorganisms living in water containing only 1000 Bq/L of $^{222}$Rn.

We, also, observe that the highest number of SSBs/Gy/Mbp and DSBs/Gy/Mbp is predicted for the benthic mixture, while the frustule decreases the values by 6%. The number of SSBs/Gy/Mbp originating solely from $^{222}$Rn in the water is 14% higher than from $^{226}$Ra in the dry sediments. When we normalize the SSBs and DSBs for 27 Mbp and 1-day exposure to 30 Bq/g of $^{226}$Ra and 1000 Bq/L of $^{222}$Rn, we observe the opposite trend; the number of SSBs/day originating from $^{222}$Rn in the water is 97% lower than from $^{226}$Ra in the dry sediments.

## Discussion

### Absorbed dose to microorganisms

The goal of this work was to apply for the first time micro- and nanodosimetric approaches and tools to evaluate the doses received by microorganisms living in naturally radioactive ecosystems. We simulated separately the $\alpha$-particles of $^{222}$Rn in the water column and $^{226}$Ra in the sediments of mineral springs in Auvergne, excluding their daughter nuclei in a first stage. Indeed, the contribution to the dose rates coming from the radioelements in the decay chain of $^{226}$Ra and $^{222}$Rn, especially the $\alpha$-emitters, depends on their location on the vicinity of the diatoms. Additional data are needed to understand their chemical behaviour and therefore their contribution to the dose rates to the microorganisms.

Diatoms were selected as our model organism because they were observed to display deformations in the most radioactive spring studied in Auvergne. Their frustule, composed of amorphous silicate, has been considered in the simulations in order to evaluate its impact on the absorbed dose rate compared to other microorganisms. We remind that for the purpose of this paper we considered only an external radiative environment, excluding the uptake of radionuclides through the pores of the frustule through which nutrients present in the water are absorbed.

As shown in Table 7, the interactions of $\alpha$-particles with the 2 μm thickness frustule, with a density 2.4 times higher than that of the water, decrease the maximum kinetic energy of the particles entering the diatom by 6% on average. The impact of the frustule is also observed in the absorbed dose rates to diatoms which are 10% lower than for other microorganisms in the same environmental conditions.

As shown in Fig 3, the energy deposited from $^{222}$Rn ($E_{\alpha,max}$ = 5.5 MeV, maximum range in water = 43.4 μm) is slightly higher than the one from $^{226}$Ra ($E_{\alpha,max}$ = 4.8 MeV, maximum range in water = 35 μm). Therefore, $\alpha$-particles will deposit a great part of their energy in the microorganism (10 μm radius) through ionisations, reaching the maximum energy deposition at the end of their range (peak observed at higher energies). Considering the same number of primaries generated for both radionuclides, the absolute dose calculated to the diatom is 26% higher for $^{222}$Rn than for $^{226}$Ra.

When dose rates are normalized to the reference activity concentrations in the environment (30 Bq/g of $^{226}$Ra in sediments and 1000 Bq/L of $^{222}$Rn in the water column), we reach 92.4 μGy/h when considering only dry sediments, 2.8 μGy/h considering only water, and 9.7 μGy/h considering both $^{222}$Rn and $^{226}$Ra contributions in the benthic mixture (see Table 8 and Fig 3). To our knowledge, dose rates to diatoms have so far been poorly documented; we could only compare to the work of Morthekai et al. [74], who investigated luminescence dating on diatom fossils in core sediments of a river and a lake. They obtained a range of values between 0.5 and 1 μGy/h due to U, Th and K. Although the environments and radionuclides differ from our work, Morthekai's values are coherent with our calculations considering that 1.1 μGy/h is deposited at the frustule in the benthic mixture.

Moreover, it is interesting to compare our dose rate values to those recommended by the ERICA risk assessment tool [6] for the protection of the environment. ERICA provides a 10 μGy/h dose rate threshold for all the ecosystems and non-human species. Below this threshold, the environmental risks are considered negligible.

This limit value is almost reached for the benthic conditions considered in our study while it is crossed for the activities measured in spring 5 (Chateldon—values shown in Tables 1 and 2). Indeed, the dose rate to benthic diatoms in spring 5 coming only from external $^{226}$Ra and $^{222}$Rn is 18.3 μGy/h. Additional exposure is expected to come from other radioelements present either outside the diatom frustule or internally incorporated [75]. The disequilibrium observed in the $^{238}$U and $^{232}$Th decay chains confirms that the dose rate calculations should not be performed under the equilibrium hypothesis but rather according to the measured activity concentrations, as adopted by ERICA.

## DNA damage in microorganisms

As presented in Table 7, whatever the considered environment, the mean energy of $\alpha$-particles reaching the nucleus is around 2.3 MeV corresponding to a range of 13 μm, 13 times higher than the nucleus diameter considered (1 μm). Consequently, a very small fraction of energy (around 0.07%) is deposited to the nucleus. When considering the frustule, the mean kinetic energy of $\alpha$-particles is reduced by 10% maximum.

As shown in Figs 4–6, at nanoscale, the dominant processes in terms of probabilities are $\alpha$-particles ionisations and excitations, as well as, ionisations, excitations and solvations of secondary electrons. Solvated electrons (free electrons in liquid water), known to play an important role in the damaging effects to the DNA [76, 77], are contributing to 15% of the total specific energy, while electrons and $\alpha$-particles ionisations contribution to the total specific energy is 32.5% and 19.6% respectively.

From Table 10, we can remark that the frustule reduces the SE rates to a nucleosome by 11% which is very coherent with the reduction of the kinetic energy observed. The highest SE rates are obtained for 0% porosity (30 Bq/g of $^{226}$Ra in the dry sediments) with a 147 bp DNA receiving 34 times higher SE rates in comparison to an environment characterized by 100% porosity (1000 Bq/L of $^{222}$Rn in the water).

For the DNA damage assessment, we first performed a validation of the proper SPointProb value to be used in DBSCAN algorithm. The algorithm has already been validated for protons using 16% [52] and 20% SPointProb [48] to fit respective experimental and simulation data. For $\alpha$-particles in the energy range of 2—10 MeV, we tested 20%, 16%, 12% and 8% SPoint-Prob in an effort to best fit our results with Moeini et al. including the literature provided [67]. As we can see in Fig 7, our values of DSB/Gy/Mbp for the chosen 8% SPointProb lie within the simulation and experimental data found in the literature. After the validation, we were able to predict that the highest number of SSBs/Gy/Mbp and DSBs/Gy/Mbp takes place in the benthic mixture (90% porosity) where the effect of the frustule is no more evident.

Table 10 displays the number of SSBs and DSBs for 1-day exposure to 30 Bq/g of $^{226}$Ra and 1000 Bq/L of $^{222}$Rn considering a 27 Mbp genome. We highlight that the number of Single DNA strand breaks can differ by 1 order of magnitude from 4.50E-03 SSB / day for 0% poros-ity to 1.48E-04 SSB / day for 100% porosity. When Lampe et al. [78] conducted similar studies for the DNA damage induced on the prokaryotic *Escherichia coli* due to the natural back-ground radiation, they observed that the natural radiation background near the surface was responsible for only 2.8E-08 DSB/day. As *E. coli* spontaneous mutation rate from endogenous causes is orders of magnitude higher (1.0E-03 / cell division) [14], they concluded that the background radiation had likely only a very small mutational effect on the biological system under study. With our work, we show that the computed mutation rate (4.7E-04 DSB/day) for the diatoms in the studied mineral springs is 4 orders of magnitude higher than for bacteria exposed solely to natural background radiation (2.8E-08 DSB/day) as calculated by Lampe et al. [78].

Recently, experiments to evaluate the spontaneous mutation rate in the model diatom *Phaeodactylum tricornutum* were conducted for the first time revealing a total spontaneous nuclear mutation rate per generation of approximately 1.29E-02 (accounting for base substitu-tion and insertion-deletion mutation rates) [79].

When comparing these numbers to the rate of radiation induced SSBs and DSBs deduced from Table 10, respectively 4.70E-04 and 1.11E-04 per generation day for the benthic mixture, we observe that radiation induced mutations could contribute to an accountable mutational pressure. Our result can be related to the observed correlation between natural bedrock radio-activity and the mutation rate of waterlices living in subterranean habitants [2].

This result suggests that natural radioactivity can be an important abiotic driver of the evo-lution of microorganisms living in mineral springs. The comparison has of course some limita-tions. First, laboratory conditions impose their own sources of stress with a potential effect on the mutation rates. As a consequence, differences are expected between the experimental mutation rates and the long-term average mutation rates in the natural environment. The other limitation lies in the comparison between radiation induced damages and spontaneous mutation rates. Between the two observables lies the complete cell repair process.

## Conclusion

In this study, we focused on the simulation of the external radiation exposure of microorgan-isms and diatoms living in naturally radioactive aquatic ecosystems. Three different environ-mental compositions were simulated corresponding to the ecosystems of mineral springs: dry

sediments (0% porosity) containing only 30 Bq/g $^{226}$Ra, water (100% porosity) containing only 1000 Bq/L $^{222}$Rn and a benthic mixture of both (90% porosity) representing realistic conditions. In the benthic mixture, the diatom is exposed to 9.7 μGy/h due to $^{226}$Ra and $^{222}$Rn, a dose rate which is comparable with the threshold (10 μGy/h) for the protection of the ecosystems suggested by ERICA risk assessment tool. We evaluated that the frustule does not considerably protect diatoms from ionizing radiation. Based on our computed DSB, we show that the microorganisms are extremely exposed to DNA damages due to the chronic exposure to ionising radiation in the radioactive mineral springs. By demonstrating the coupling of experimental measurements with Monte Carlo simulations for two radionuclides, this work can be implemented in future radioecological studies wishing to estimate not only dose rates but also potential DNA damages on aquatic microorganisms and extend to other radioisotopes.

## Acknowledgments

Computations have been performed on the supercomputer facilities of the Mésocentre Clermont Auvergne University. We acknowledge the valuable support of GEANT4-DNA collaboration, as well as, the fruitful discussions with Theo J. Mertzimekis. We would like also to thank Alexis Pereda for his contribution to the software archive preparations. The other members of the TIRAMISU collaboration are: Elisabeth Allain (Laboratoire de Géographie Physique et Environnementale (GEOLAB)—UMR6042, CNRS, Université Clermont Auvergne, Clermont Ferrand, France), Sylvia Becerra (Laboratoire Géosciences Environnement Toulouse (LaSSP)—UMR5563, CNRS, Université Toulon, Toulon, France), Emmanuel Busato (Laboratoire de Physique de Clermont (LPC)—UMR6533, CNRS/IN2P3 Université Clermont Auvergne, Aubière, France), Helene Celle (Laboratoire Chrono-Environnement UMR6249, CNRS Université de Bourgogne Franche-Comté, Besançon, France), Jonathan Colombet (Laboratoire Microorganismes: Génome Environnement (LMGE)—UMR6023, CNRS, Université Clermont Auvergne, Clermont Ferrand, France), Pierre-Jean Gauthier (Laboratoire Magmas et Volcans (LMV)—UMR6524, CNRS/INSU Université Clermont Auvergne, Aubiere, France), Yihua He (Laboratoire de physique subatomique et des technologies associées (SUBATECH)—UMR6457, CNRS/IN2P3/IMT Atlantique/Université de Nantes, Nantes, France), Guillaume Holub (Laboratoire de Physique des 2 infinis de Bordeaux (LP2iB)—UMR5707, CNRS/IN2P3/CENBG Université de Bordeaux, Gradignan, France), Anne-Helene Le Jeune (Laboratoire Microorganismes: Génome Environnement (LMGE)—UMR6023, CNRS, Université Clermont Auvergne, Clermont Ferrand), Clarisse Mallet (Laboratoire Microorganismes: Génome Environnement (LMGE)—UMR6023, CNRS, Université Clermont Auvergne, Clermont Ferrand, France), Justine Marchand (Metabolism, Molecular Engineering of Microalgae and Applications, Laboratoire de Biologie des Organismes, Stress, Santé Environnement, IUML FR3473, CNRS, Le Mans Université, Le Mans, France), Herve Michel (Institut de Chimie de Nice (ICN)—UMR7272, CNRS Université Côte d'Azur, Nice, France), Olivier Peron (Laboratoire de physique subatomique et des technologies associées (SUBATECH)—UMR6457, CNRS/IN2P3/IMT Atlantique/Université de Nantes, Nantes, France), Pradeep Ram Angia Sriram (Laboratoire Microorganismes: Génome Environnement (LMGE)—UMR6023, CNRS, Université Clermont Auvergne, Clermont Ferrand, France), Claire Sergeant (Laboratoire de Physique des 2 infinis de Bordeaux (LP2iB)—UMR5707, CNRS/IN2P3/CENBG Université de Bordeaux, Gradignan, France), Marie-Helene Vesvres (Laboratoire de Physique des 2 infinis de Bordeaux (LP2iB)—UMR5707, CNRS/IN2P3/CENBG Université de Bordeaux, Gradignan, France), Olivier Voldoire (Laboratoire de Géographie Physique et Environnementale (GEOLAB)—UMR6042, CNRS, Université Clermont Auvergne, Clermont Ferrand, France).

## Author Contributions

**Conceptualization:** Sofia Kolovi, Vincent Breton, Lydia Maigne.

**Data curation:** Sofia Kolovi, Patrick Chardon, Didier Miallier, Vincent Breton, Lydia Maigne.

**Formal analysis:** Sofia Kolovi, Giovanna-Rosa Fois, Sarra Lanouar, Vincent Breton, Lydia Maigne.

**Funding acquisition:** Aude Beauger, David G. Biron, Gilles Montavon, Vincent Breton, Lydia Maigne.

**Investigation:** Sofia Kolovi, Giovanna-Rosa Fois, Sarra Lanouar, Patrick Chardon, Didier Miallier, Lory-Anne Baker, Céline Bailly, Aude Beauger, David G. Biron, Karine David, Gilles Montavon, Thierry Pilleyre, Benoît Schoefs, Vincent Breton, Lydia Maigne.

**Methodology:** Sofia Kolovi, Giovanna-Rosa Fois, Vincent Breton, Lydia Maigne.

**Project administration:** Aude Beauger, David G. Biron, Vincent Breton, Lydia Maigne.

**Resources:** Didier Miallier, Thierry Pilleyre, Vincent Breton, Lydia Maigne.

**Software:** Sofia Kolovi, Giovanna-Rosa Fois, Sarra Lanouar, Lydia Maigne.

**Supervision:** Vincent Breton, Lydia Maigne.

**Validation:** Sofia Kolovi, Giovanna-Rosa Fois, Patrick Chardon, Vincent Breton, Lydia Maigne.

**Visualization:** Sofia Kolovi.

**Writing – original draft:** Sofia Kolovi, Giovanna-Rosa Fois, Patrick Chardon, Vincent Breton, Lydia Maigne.

**Writing – review & editing:** Sofia Kolovi, Giovanna-Rosa Fois, Sarra Lanouar, Patrick Chardon, Didier Miallier, Lory-Anne Baker, Céline Bailly, Aude Beauger, David G. Biron, Karine David, Gilles Montavon, Thierry Pilleyre, Benoît Schoefs, Vincent Breton, Lydia Maigne.

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
