## [Decision Letter · Decision Letter 0]

8 Jan 2023

PONE-D-22-32535Simulation of the radiation exposure of microorganisms living in naturally radioactive mineral springs using GATE and Geant4-DNA Monte Carlo simulation toolsPLOS ONE

Dear Dr. Kolovi,

Thank you for submitting your manuscript to PLOS ONE. After careful consideration, we feel that it has merit but does not fully meet PLOS ONE’s publication criteria as it currently stands. Therefore, we invite you to submit a revised version of the manuscript that addresses the points raised during the review process.

We look forward to receiving your revised manuscript.

Kind regards,

Mohamad Syazwan Mohd Sanusi

Academic Editor

PLOS ONE

Journal Requirements:

3. Please note that PLOS ONE has specific guidelines on code sharing for submissions in which author-generated code underpins the findings in the manuscript. In these cases, all author-generated code must be made available without restrictions upon publication of the work. 

Please review our guidelines at https://journals.plos.org/plosone/s/materials-and-software-sharing#loc-sharing-code and ensure that your code is shared in a way that follows best practice and facilitates reproducibility and reuse.

5. Please expand the acronym “CNRS” (as indicated in your financial disclosure) so that it states the name of your funders in full.

6. Thank you for stating the following financial disclosure: 

"This study was funded by "Prime80 CNRS", contract No 1083577"

7. In your Data Availability statement, you have not specified where the minimal data set underlying the results described in your manuscript can be found. PLOS defines a study's minimal data set as the underlying data used to reach the conclusions drawn in the manuscript and any additional data required to replicate the reported study findings in their entirety. All PLOS journals require that the minimal data set be made fully available. For more information about our data policy, please see http://journals.plos.org/plosone/s/data-availability.

8. One of the noted authors is a group or consortium:  TIRAMISU collaboration 

In addition to naming the author group, please list the individual authors and affiliations within this group in the acknowledgments section of your manuscript. Please also indicate clearly a lead author for this group along with a contact email address.

Reviewers' comments:

Reviewer's Responses to Questions

**Comments to the Author**

1. Is the manuscript technically sound, and do the data support the conclusions?

Reviewer #1: Yes

Reviewer #2: Yes

2. Has the statistical analysis been performed appropriately and rigorously? 

Reviewer #1: Yes

Reviewer #2: Yes

3. Have the authors made all data underlying the findings in their manuscript fully available?

Reviewer #1: Yes

Reviewer #2: Yes

4. Is the manuscript presented in an intelligible fashion and written in standard English?

Reviewer #1: Yes

Reviewer #2: Yes

5. Review Comments to the Author

Reviewer #1: The manuscript presents calculations of absorbed dose and dose rate to microorganisms living in mineral springs with high abundance of naturally occurring radioactive elements. The calculations are based on measured activity concentration and performed by available program code for Monte Carlo simulations. The manuscript is well written and structured but a few issues need to be addressed.

p.4: Water was collected in beakers that were sealed to avoid leakage of Rn-222. It should be described how the sampling was performed to avoid exhalation of radon gas during the sampling.

p.4: It is not obvious if the contribution to absorbed dose from all alpha emitters in the uranium and thorium decay chains is considered, or if only the contribution from Rn-222 and Ra-226 is considered. This is important to discuss in the discussion part, especially when making comparisons with the ERICA tool.

Table 2: Neither the uranium nor the thorium decay chains seem to be in equilibrium. Instead there seems to be an enrichment of Ra at most of the studied sites. The reason for this ought to be explained. The degree of disequilibrium will also affect the absorbed dose to organisms in these environments, if all alpha emitters are considered.

Reviewer #2: I have had an opportunity to review an interesting submission with a title “Simulation of the radiation exposure of microorganisms living in naturally radioactive mineral springs using GATE and Geant4-DNA Monte Carlo simulation tools” by Sofia Kolovi et al.

In general, the paper is well thought out and well written, in a concise and understandable English. Despite that, I have found few things in the running text, mentioned further below in my comments, which I think needs attention.

In the article the authors have taken samples from five different natural mineral springs in France, and performed radiological characterisation of these springs. With the calculated activity concentration values, they then tried to mimic the conditions within the springs in a Geant4 model at three positions – sediment position, mixed position and water position, replicating radiological conditions at these positions in the springs. Then they created a semi-detailed diatom algae model out of spheres of water, and tried to estimate the amount of single and double strand breaks in the DNA of the diatom algae.

The results are interesting, as they show that the number of double strand brakes, which is indirectly correlated with the number of mutations of the DNA, can be expected to be high in such mineral spring conditions – particularly spring five – than as for other microorganisms in normal radiation background environments. It would be interesting to see this applied to help solving the human low-dose radiation response problem in radiation protection by trying similar studies for human tissues.

I recommend a minor revision.

Below you will find my comments to this manuscript:

1. Line 151. I think it would be helpful to at least mention what is considered to be the deformation, not just leaving a reference to another paper. I think, lightly incorporating a brief definition of a deformation would be very helpful for the readers.

2. Line 188. “were emitted isotropically” – I would say being a bit more detailed than isotropically would be very helpful for the clarity of the text. Something along the lines of “the modelled radioactive nuclei were distributed randomly across material X and with the directions random in the solid angle of X π…“ would be way more clearer for the reader.

3. In Table 4, a list of energies of α particles is provided. Detail is lacking regarding how they were simulated and calculated. In the running text it is mentioned that the total energy deposited was counted, but later in results, the results for 226Ra and 222Rn are separated. Could you please clarify this a bit more?

4. Line 269. Am I misunderstanding something, or is “ion ionisation” a typo? Can you really ionise an ion?

6. In Table 10, the values displayed in the parentheses are for diatoms with frustules. This needs to be mentioned explicitly above or below the table. Otherwise it can get confusing for the reader, and it takes unnecessary time to work this out.

7. Line 372. It would be nice if the “solvations of secondary electrons” would be very briefly explained to avoid confusion, as the immediate-previous context of this phrase within the running text discusses physics of radiation interaction with matter.

8. Figure 6 – There is basically no discussion regarding the comparison of the results of the article to the other studies, apart from saying that the values are in “good agreement”. I think, a bit more thorough comparison – with at least one of the other studies – would be very beneficial to further strengthening the paper. Additionally, it would be very nice to see how the error bars look on all of the SpointProb graphs.

6. PLOS authors have the option to publish the peer review history of their article (what does this mean?). If published, this will include your full peer review and any attached files.

Reviewer #1: No

Reviewer #2: No

---

## [Author Response · Author response to Decision Letter 0]

24 Jan 2023

Dear editor, 

We would like to express our gratitude for the interest expressed for our manuscript. We highly appreciate the relevance of all the comments made by the reviewers and your additional guidance to meet the journal’s requirements. 

Please find below our answers to the reviewer comments as well as a proposed revision of the manuscript. We hope this revised version will improve its quality, meet the journal’s criteria, and merit publication.

Kind regards,

Kolovi Sofia on behalf of the authors

Journal Requirements:

Requirement 1. Please ensure that your manuscript meets PLOS ONE's style requirements, including those for file naming. The PLOS ONE style templates can be found at 

Response to Requirement 1: The manuscript has been revised according to the plos_latex_template_v3.6 (https://journals.plos.org/plosone/s/latex).

Requirement 2. In your Methods section, please provide additional information regarding the permits you obtained for the work. Please ensure you have included the full name of the authority that approved the field site access and, if no permits were required, a brief statement explaining why.

Response to Requirement 2: No authority permissions are required for the studied mineral springs since they belong to the public sector with no restrictions applied. In order to clarify this point, additional text was added in Line 100: “The springs are open for public use and no permit was required to access them and collect samples”. 

Requirement 3. Please note that PLOS ONE has specific guidelines on code sharing for submissions in which author-generated code underpins the findings in the manuscript. In these cases, all author-generated code must be made available without restrictions upon publication of the work. 

Please review our guidelines at https://journals.plos.org/plosone/s/materials-and-software-sharing#loc-sharing-code and ensure that your code is shared in a way that follows best practice and facilitates reproducibility and reuse.

Response to Requirement 3: The code can be found in https://github.com/lpc-umr6533/tiramisu_simulation and it will be immediate available to the public once required by the journal.

Requirement 4. We note that the grant information you provided in the ‘Funding Information’ and ‘Financial Disclosure’ sections do not match. 

Response to Requirement 4: We modified the “Funding Information” section accordingly.

Requirement 5. Please expand the acronym “CNRS” (as indicated in your financial disclosure) so that it states the name of your funders in full.

Response to Requirement 5: Centre National de la Recherche Scientifique (CNRS)

Requirement 6. Thank you for stating the following financial disclosure: 

"This study was funded by "Prime80 CNRS", contract No 1083577"

Response to Requirement 6: The text in the financial disclosure has been modified as following: "The current study was performed under S. Kolovi’s PhD funding received by the Centre National de la Recherche Scientifique (CNRS) as a "Prime80 CNRS" project with contract No 1083577. The funders had no role in study design, data collection and analysis, decision to publish, or preparation of the manuscript." 

Requirement 7. In your Data Availability statement, you have not specified where the minimal data set underlying the results described in your manuscript can be found. PLOS defines a study's minimal data set as the underlying data used to reach the conclusions drawn in the manuscript and any additional data required to replicate the reported study findings in their entirety. All PLOS journals require that the minimal data set be made fully available. For more information about our data policy, please see http://journals.plos.org/plosone/s/data-availability.

Response to Requirement 7: We modified the Data Availability statement as it follows: “The simulation code is available on GitHub (https://github.com/lpc-umr6533/tiramisu_simulation). All relevant parameters and data are within the manuscript, allowing the reproducibility of the results.”

Requirement 8. One of the noted authors is a group or consortium: TIRAMISU collaboration 

In addition to naming the author group, please list the individual authors and affiliations within this group in the acknowledgments section of your manuscript. Please also indicate clearly a lead author for this group along with a contact email address.

Response to Requirement 8: The affiliations of the authors are added in the acknowledgements section. The lead author of the collaboration is among the co-authors of the article and his email address has been communicated accordingly below the authors list. 

Requirement 9. Please review your reference list to ensure that it is complete and correct. If you have cited papers that have been retracted, please include the rationale for doing so in the manuscript text, or remove these references and replace them with relevant current references. Any changes to the reference list should be mentioned in the rebuttal letter that accompanies your revised manuscript. If you need to cite a retracted article, indicate the article’s retracted status in the References list and also include a citation and full reference for the retraction notice.

Response to Requirement 9: One reference was removed because the article is still under revision and it has not been accepted yet (Line 68: Baker LA, et al.). Additional references were added, and the reference list was accordingly reviewed. Both the removed and added references are stated at the end of the rebuttal letter.

Response to Reviewer #1

Point 0 - The manuscript presents calculations of absorbed dose and dose rate to microorganisms living in mineral springs with high abundance of naturally occurring radioactive elements. The calculations are based on measured activity concentration and performed by available program code for Monte Carlo simulations. The manuscript is well written and structured but a few issues need to be addressed.

Response to Point 0 – We are very thankful for the interest expressed by the reviewer and the very useful comments which highlighted important aspects of our work. The points raised are addressed below.

Point 1, p.4 - Water was collected in beakers that were sealed to avoid leakage of Rn-222. It should be described how the sampling was performed to avoid exhalation of radon gas during the sampling.

Response to Point 1 - Water sampling was performed according to ISO 5667-1 standard for water sampling and ISO 5667-3 standard concerning water sample handling and preservation for radiochemical analysis. We used Marinelli style gas analysis containers from NUVIA Instruments GMBH that are specifically designed for γ-spectroscopy analyses of gases including radon. We added this information in the text:

Lines 111 - 113: “Water was collected according to ISO 5567-1 and ISO 5567-3 standards in 1 L Marinelli style gas analysis containers (NUVIA Instruments GMBH) designed for γ-spectroscopic analysis. The containers were sealed to avoid any 222Rn leakage.”

Point 2, p.4 - It is not obvious if the contribution to absorbed dose from all alpha emitters in the uranium and thorium decay chains is considered, or if only the contribution from Rn-222 and Ra-226 is considered. This is important to discuss in the discussion part, especially when making comparisons with the ERICA tool.

Table 2: Neither the uranium nor the thorium decay chains seem to be in equilibrium. Instead there seems to be an enrichment of Ra at most of the studied sites. The reason for this ought to be explained. The degree of disequilibrium will also affect the absorbed dose to organisms in these environments, if all alpha emitters are considered.

Response to Point 2: We would like to thank the reviewer for this highly important remark. It is now clarified in the manuscript that only the contribution of 222Rn and 226Ra is considered. Indeed, active hydrothermal circulation affects numerous areas of Massif Central. A common feature of several CO2-rich geothermal systems is a high radium content acquired during the ascent of water from the deep reservoir through the crystalline bedrock. High radium solubility in the mineral waters can be related to the major ion composition [21]. The following text was added:

Lines 127 - 133: “A common feature of several CO2-rich geothermal systems is a high radium content acquired during the ascent of water from the deep reservoir through the crystalline bedrock. High radium solubility in the mineral waters can be related to their major ion composition [21]. As a consequence, the sediments and deposits (travertine) precipitated from the waters display a disequilibrium in the uranium and thorium decay chains.”

The text between Lines 197 - 199 was also modified accordingly: “Following the reduced levels of activities measured for 238U, 228Ra and 228Th (see Table 2) in comparison to 222Rn and 226Ra, as well as the disequilibrium of the decay chains observed in the sediments, we simulated only α-particles emitted by the latter without considering their daughter nuclei.”

In the Discussion part, text was added in Lines 357 – 359: “The simulations were performed separately for the α-particles of 222Rn and 226Ra, excluding their daughter nuclei.”

Lines 393 – 396: “The disequilibrium observed in the 238U and 232Th decay chains confirms that the dose rate calculations should not be performed under the equilibrium hypothesis but rather according to the measured activity concentrations, as adopted by ERICA.”

Response to Reviewer #2

Point 0 - I have had an opportunity to review an interesting submission with a title “Simulation of the radiation exposure of microorganisms living in naturally radioactive mineral springs using GATE and Geant4-DNA Monte Carlo simulation tools” by Sofia Kolovi et al.

In general, the paper is well thought out and well written, in a concise and understandable English. Despite that, I have found few things in the running text, mentioned further below in my comments, which I think needs attention.

In the article the authors have taken samples from five different natural mineral springs in France, and performed radiological characterisation of these springs. With the calculated activity concentration values, they then tried to mimic the conditions within the springs in a Geant4 model at three positions – sediment position, mixed position and water position, replicating radiological conditions at these positions in the springs. Then they created a semi-detailed diatom algae model out of spheres of water, and tried to estimate the amount of single and double strand breaks in the DNA of the diatom algae.

The results are interesting, as they show that the number of double strand brakes, which is indirectly correlated with the number of mutations of the DNA, can be expected to be high in such mineral spring conditions – particularly spring five – than as for other microorganisms in normal radiation background environments. It would be interesting to see this applied to help solving the human low-dose radiation response problem in radiation protection by trying similar studies for human tissues.

I recommend a minor revision.

Response to Point 0 – We would like to thank the reviewer for their interest and the encouraging review. The comments were all well taken into serious consideration and respective modifications in the text were applied. 

Below you will find my comments to this manuscript:

Point 1, Line 151 - I think it would be helpful to at least mention what is considered to be the deformation, not just leaving a reference to another paper. I think, lightly incorporating a brief definition of a deformation would be very helpful for the readers.

Response to Point 1 – The corresponding text has been supplemented by the description of the deformations considered in reference [35] as shown below:

Lines 161 - 164: “The same species are observed at spring 5 (Chateldon) where deformation rates above 25%, denoted by abnormally shaped frustules, have been observed on Planothidium frequentissimum [35]. Diatom deformations or else stated “teratological formations” are abnormalities on their morphology, mainly on the valve shape and structural characteristics which result in deformed frustules.”

Point 2, Line 188 - “were emitted isotropically” – I would say being a bit more detailed than isotropically would be very helpful for the clarity of the text. Something along the lines of “the modelled radioactive nuclei were distributed randomly across material X and with the directions random in the solid angle of X π…“ would be way more clearer for the reader.

Response to Point 2 - The suggestion has been implemented in the text as it follows:

Lines 202 – 205: “For each radionuclide, the α-particles were emitted isotropically (4π solid angle) from the spherical volume surrounding the microorganism where their emission point was randomly distributed.” 

Point 3 - In Table 4, a list of energies of α-particles is provided. Detail is lacking regarding how they were simulated and calculated. In the running text it is mentioned that the total energy deposited was counted, but later in results, the results for 226Ra and 222Rn are separated. Could you please clarify this a bit more?

Response to Point 3 - References1,2 were added in Table 4 legend pointing to the databases from which the α-particles energies were retrieved. Additional text was inserted in order to clarify that the two radionuclides were simulated separately: 

Line 205: “Separate simulations were performed for 222Rn and 226Ra.”

Lines 223 - 224: “We ran separate simulations with 10E+08 α-particles for each radionuclide and repeated them 10 times to evaluate the statistical fluctuations (kept below 1%).”

Regarding the “total energy deposited”, indeed, the energy deposited by each radionuclide was recorded and then, the total energy deposited was also calculated. We modified the text accordingly:

Lines 218 - 220: “Then, the energy deposited to the microorganism for each radionuclide was recorded, the contributions of the dominant physical processes were identified, and the total energy depositions were calculated.”

Point 4, Line 269 - Am I misunderstanding something, or is “ion ionisation” a typo? Can you really ionise an ion?

Response to Point 4 – We would like to thank the reviewer for this very critical remark. Indeed, we kept the terminology from Geant4 concerning the mechanisms describing energy depositions. The alpha particle is classified as an ion in the MC code since it is a double-ionised He atom. When the alpha particle ionizes the matter, the process responsible for the energy deposited is “IonIonisation”. 

As long as the alpha particle is able to produce secondary electrons able to further ionise the medium (called δ-rays), the energy depositions of these secondary electrons are denoted as “electron ionisations”. In order to clarify this for the reader, the following text was added: 

Lines 289 – 291: “Ion ionisation refers to the ionisations caused directly from the α-particle while the electron ionisation refers to the ionisations caused by sufficiently energetic secondary electrons (δ-rays).”

Point 6 - In Table 10, the values displayed in the parentheses are for diatoms with frustules. This needs to be mentioned explicitly above or below the table. Otherwise it can get confusing for the reader, and it takes unnecessary time to work this out.

Response to Point 6 – We thank the reviewer for highlighting this potential point of confusion for the reader. We added the following part in the legend of Table 10: 

“Table 10. Mean specific energy (SE) rates per nucleosome (μGy/h) and number of SSBs, DSBs (when considering frustule, values are provided in the parenthesis)”

Point 7, Line 372 - It would be nice if the “solvations of secondary electrons” would be very briefly explained to avoid confusion, as the immediate-previous context of this phrase within the running text discusses physics of radiation interaction with matter.

Response to Point 7 – The following text was added: 

Line 405: “Solvated electrons (free electrons in liquid water) known to play an important role in the damaging effects of DNA [73,74] to…”

Point 8, Figure 6 – There is basically no discussion regarding the comparison of the results of the article to the other studies, apart from saying that the values are in “good agreement”. I think, a bit more thorough comparison – with at least one of the other studies – would be very beneficial to further strengthening the paper. Additionally, it would be very nice to see how the error bars look on all of the SpointProb graphs.

Response to Point 8, Figure 6 – We understand the concern raised by the plot. Its discussion is detailed later in the paper (lines 415 - 421). We modified the text as it follows: 

Lines 322 - 327: “As it is shown, our predicted values are in general agreement with the experimental and other simulation results using different MC codes. The overprediction of DSBs when using the suggested values of 16% [53] and 20% [49] smooths when lowering the value. 12% offers a good agreement with H. Moeini et al. [64] but we finally chose SpointProb equal to 8% due to the better agreement with the experimental results, too.”

We hope that this modification fits well with the Discussion lines 415 - 421, where we provide the full explanation of the impact of the modification of SPointProb values on the results.

In order to evaluate the error bars, only due to the statistics, in Fig 6, we ran the simulation multiple times and we achieved a relative uncertainty below 0.1% (too small to be visible in the graph). For clarification, the following text was added:

Line 275: “We ran the simulation multiple times, achieving a relative uncertainty below 0.1%.”

ADDITIONAL REFERENCES 

1. Singh, S., Jain, A. K. & Tuli, J. K. Nuclear Data Sheets for A = 222. Nucl. Data Sheets 112, 2851–2886 (2011).

2. Singh, B. et al. Nuclear Data Sheets for A=218. Nucl. Data Sheets 160, 405–471 (2019).

REMOVED REFERENCES

From line 68: 

Baker L.A., Beauger A., Kolovi S., Voldoire O., Allain E., Breton V., Chardon P., Miallier D., Bailly C., Montavon G., Bouchez A., Rimet F., Chardon C., Vasselon V., Ector L., Wetzel C.E., Biron D.G. Diatom DNA metabarcoding to assess the effect of natural radioactivity in mineral springs on genetic variants of benthic diatoms communities. Submitted to Science of the total environment. September 2002.

---

## [Decision Letter · Decision Letter 1]

5 Apr 2023

PONE-D-22-32535R1Simulation of the radiation exposure of microorganisms living in naturally radioactive mineral springs using GATE and Geant4-DNA Monte Carlo simulation toolsPLOS ONE

Dear Dr. Sofia Kolovi,

Thank you for submitting your manuscript to PLOS ONE. After careful consideration, we feel that it has merit but does not fully meet PLOS ONE’s publication criteria as it currently stands. Therefore, we invite you to submit a revised version of the manuscript that addresses the points raised during the review process.

We look forward to receiving your revised manuscript.

Kind regards,

Mohamad Syazwan Mohd Sanusi

Academic Editor

PLOS ONE

Journal Requirements:

Reviewers' comments:

Reviewer's Responses to Questions

**Comments to the Author**

1. If the authors have adequately addressed your comments raised in a previous round of review and you feel that this manuscript is now acceptable for publication, you may indicate that here to bypass the “Comments to the Author” section, enter your conflict of interest statement in the “Confidential to Editor” section, and submit your "Accept" recommendation.

Reviewer #2: All comments have been addressed

Reviewer #3: (No Response)

Reviewer #4: (No Response)

2. Is the manuscript technically sound, and do the data support the conclusions?

Reviewer #2: Yes

Reviewer #3: Yes

Reviewer #4: Yes

3. Has the statistical analysis been performed appropriately and rigorously? 

Reviewer #2: Yes

Reviewer #3: Yes

Reviewer #4: Yes

4. Have the authors made all data underlying the findings in their manuscript fully available?

Reviewer #2: Yes

Reviewer #3: Yes

Reviewer #4: Yes

5. Is the manuscript presented in an intelligible fashion and written in standard English?

Reviewer #2: Yes

Reviewer #3: Yes

Reviewer #4: Yes

6. Review Comments to the Author

Reviewer #2: I am happy in the way the comments have been addressed, and the text of manuscript altered.

I have no further comments.

Reviewer #3: Overall: This is an extremely interesting topic. This multi-disciplinary study appears to have been planned and executed in accordance with standards and accepted experimental protocols.

Introduction: This section is well written with a clean layout. Readers will appreciate the step-by-step introduction of the finer points of the experiment motivation.

Line 87: Briefly define and describe the DBSCAN algorithm.

Materials and methods: I appreciated the detailed description of the sampling methods, protocols, and standards followed. The Geant4 geometry was modeled appropriately for the scenarios and described effectively.

Table 1: It might be useful to show a map showing the sample locations of the springs amongst the macroscopic bodies of water in the area for readers unfamiliar with the location.

Table 2: I think +/- symbols may be missing in the numeric columns in this PDF version?

Line 235: Please explain why pure water is the appropriate material choice for the nucleosome cylinders in the model? I am left wondering why a mixture of DNA elements were not included at their respective average abundances within the nucleosome volumes.

Results:

Table 9: The use of the values in parentheses was a bit confusing at first but could be deciphered from the text above with some additional effort. Recommend adding similar clarification as is included in the caption of Table 10 (“when considering frustule, values are provided in the parenthesis”) to the caption of Table 9 as well. It is curious that the mean and maximum energies were reduced but the mean deposited energy was not? Perhaps include further explanation on this finding.

Figure 5: Define axis units in text or caption

Discussion/Conclusion:

Line 300-301,356-357,468: Line 300-301 and Figure 2 show a 10% decrease in dose rate (10.8 to 9.7 uGy/h) with frustule. Table 8 shows only one absolute dose decrease with frustule of over 20%, while the other absolute dose and dose rate deltas are between 8% and 17%. Where is the data supporting 20% dose decrease with frustule listed in lines 356-357 and 468? Please augment discussion to clarify.

Reviewer #4: This work is the first endeavor to utilize multiscale Monte-Carlo simulations to evaluate the radiation exposure of microorganisms existing in a naturally radioactive environment from α-emitters including 226Ra and 222Rn. It is innovative and provides very valuable research results and experimental data and has the potential to be employed in upcoming radioecological investigations that intend to evaluate both radiation dose rates and potential DNA damages to aquatic microorganisms and extend to other radioisotopes. The manuscript is well-written.

Point 1,

Lines 66 – 69: “An exceptional abundance of deformations in the most radioactive springs in Auvergne has been recently revealed, initiating studies of the effects of natural radioactivity on benthic diatom communities in 16 mineral springs of the

area.”

Out of 16 mineral springs, 5 were chosen. What is the reason for choosing these 5 springs? Is it according to the degree of activity concentrations of 222Rn in water?

Point 2,

In Table 2, why the sampling date for Chateldon mineral spring is March 2017, not a specific date as those for other mineral spring? Is it because that the measured mass activities of radionuclides for Chateldon mineral spring is an average of multiple measurements in March?

7. PLOS authors have the option to publish the peer review history of their article (what does this mean?). If published, this will include your full peer review and any attached files.

Reviewer #2: No

Reviewer #3: No

Reviewer #4: No

---

## [Author Response · Author response to Decision Letter 1]

25 May 2023

Dear editor, 

We would like to express our gratitude for the interest expressed for our manuscript. We highly appreciate the relevance of all the comments made by the reviewers.

Please find below our answers to the reviewer comments as well as a proposed revision of the manuscript. We performed two corrections due to incorrect data processing in the manuscript regarding the comparison of our results with the publication of Krasovec et al. as indicated below:

Lines 440 - 444: “Recently, experiments to evaluate the spontaneous mutation rate in the model diatom Phaeodactylum tricornutum were conducted for the first time revealing a total 

nuclear mutation rate of about 5.00E-10 per nucleotide per generation (accounting for 

base substitution and insertion-deletion mutation rates)” 

by 

“Recently, experiments to evaluate the spontaneous mutation rate in the model diatom Phaeodactylum tricornutum were conducted for the first time revealing a total spontaneous nuclear mutation rate per generation of approximately 1.29E−02 (accounting for base substitution and insertion-deletion mutation rates)”

Lines 445 - 448: “When comparing these numbers to the rate of radiation induced SSBs and DSBs deduced from Table 10, respectively about 0.174E-10 and 0.04E-10 per nucleotide per day, for the benthic mixture, we observe that radiation induced mutations contribute to an accountable mutational pressure.”

by

“When comparing these numbers to the rate of radiation induced SSBs and DSBs deduced from Table 10, respectively 4.70E-04 and 1.11E-04 per generation day for the benthic mixture, we observe that radiation induced mutations could contribute to an accountable mutational pressure.”

We hope this revised version will improve its quality, meet the journal’s criteria, and merit publication.

Kind regards,

Kolovi Sofia on behalf of the authors

Journal Requirements:

Requirement 1. Please review your reference list to ensure that it is complete and correct. If you have cited papers that have been retracted, please include the rationale for doing so in the manuscript text, or remove these references and replace them with relevant current references. Any changes to the reference list should be mentioned in the rebuttal letter that accompanies your revised manuscript. If you need to cite a retracted article, indicate the article’s retracted status in the References list and also include a citation and full reference for the retraction notice.

Response to Requirement 1: Two references were added in compliance with extra material asked by the reviewers.

- d-maps. Available from: https://d-maps.com/index.php?lang=en

- NIST National Institut of Standards and Technology Materials Database.

Available from: https://materialsdata.nist.gov/.

Response to Reviewer #2

Point 0 - I am happy in the way the comments have been addressed, and the text of manuscript altered. I have no further comments.

Response to Point 0 – We are grateful that our revision met the reviewer’s expectations.

Response to Reviewer #3

Point 0 - Overall: This is an extremely interesting topic. This multi-disciplinary study appears to have been planned and executed in accordance with standards and accepted experimental protocols.

Introduction: This section is well written with a clean layout. Readers will appreciate the step-by-step introduction of the finer points of the experiment motivation.

Response to Point 0 – We are very thankful for the interest expressed by the reviewer and the useful comments. The points raised are addressed below.

Point 1, Line 87 - Briefly define and describe the DBSCAN algorithm.

Response to Point 1 – The following text was added in lines 87 – 92:

DBSCAN (Density Based Spatial Clustering of Applications with Noise). For the calculation of DNA damage, DBSCAN takes into account the distribution of deposited energy induced by ionising radiation (α-particles in our case) in micrometric geometries and a damage probability function which depends on the total deposited energy.

Materials and methods: I appreciated the detailed description of the sampling methods, protocols, and standards followed. The Geant4 geometry was modeled appropriately for the scenarios and described effectively.

Point 2, Table 1 - It might be useful to show a map showing the sample locations of the springs amongst the macroscopic bodies of water in the area for readers unfamiliar with the location.

Response to Point 2 – A map (Figure 1) was added to the main body of the manuscript along with images of three of the mineral springs.

Point 3, Table 2 - I think +/- symbols may be missing in the numeric columns in this PDF version?

Response to Point 3 – Table 2 was modified accordingly.

Point 4, Line 235 - Please explain why pure water is the appropriate material choice for the nucleosome cylinders in the model? I am left wondering why a mixture of DNA elements were not included at their respective average abundances within the nucleosome volumes.

Response to Point 4 – This very interesting point is one of the major research topics of Geant4-DNA collaboration. Currently, the cross sections available in Geant4-DNA are validated for water (as defined by NIST). The following text was added in:

lines 217 – 218: G4_WATER was used as material in all cases.

lines 243 – 244: All G4-DNA models are validated and available for water (G4 WATER as defined in NIST) which is used as a surrogate to the biological medium.

Results:

Point 5, Table 9 - The use of the values in parentheses was a bit confusing at first but could be deciphered from the text above with some additional effort. Recommend adding similar clarification as is included in the caption of Table 10 (“when considering frustule, values are provided in the parenthesis”) to the caption of Table 9 as well. It is curious that the mean and maximum energies were reduced but the mean deposited energy was not? Perhaps include further explanation on this finding.

Response to Point 5 – We appreciate a lot this concern. For clarification purposes, we added the indicated text in the captions of Tables 7, 8 and 9.

(“when considering frustule, values are provided in the parentheses”)

Considering the non-obvious decrease of the mean deposited energy, we added 2 decimals in the respective values of Table 9. We would like to specify that in Table 9, we are providing only the mean and maximum kinetic energies, not the whole energy spectrum. As a result, conclusions about the relation between the reduced kinetic energy of the particles reaching the nucleus and the energy deposition at the nucleus cannot be straightforward drawn. The following text was also added in:

lines 318 – 319: We also notice that the frustule has a very limited impact on the deposited energy.

Point 6, Figure 5 - Define axis units in text or caption

Response to Point 6 – The unit of y-axis (Gy-1) was added in the captions of Figures 4, 5 and 6.

Discussion/Conclusion:

Point 7, Line 300-301,356-357,468 - Line 300-301 and Figure 2 show a 10% decrease in dose rate (10.8 to 9.7 uGy/h) with frustule. Table 8 shows only one absolute dose decrease with frustule of over 20%, while the other absolute dose and dose rate deltas are between 8% and 17%. Where is the data supporting 20% dose decrease with frustule listed in lines 356-357 and 468? Please augment discussion to clarify.

Response to Point 7 – We would like to thank the reviewer for pointing out this discrepancy between the results shown in the manuscript and the discussion part. First, we removed the respective text from line 365 where we discuss about the 20% decrease in the number of α-particles reaching the diatoms. 

Secondly, we would like to justify our calculations. In Table 8, the absolute doses in the benthic mixture are decreased due to the presence of the frustule by 22% for 226Ra and by 17.1% for 222Rn. As a result, we indicate an average 20% decrease of absolute doses.

For clarification, we modified the text in line 478 as follows:

line 478: “We showed that the main contributor to the doses is 226Ra and that the frustule is responsible for an average 20% absolute dose decrease”

Response to Reviewer #4

Point 0 - This work is the first endeavor to utilize multiscale Monte-Carlo simulations to evaluate the radiation exposure of microorganisms existing in a naturally radioactive environment from α-emitters including 226Ra and 222Rn. It is innovative and provides very valuable research results and experimental data and has the potential to be employed in upcoming radioecological investigations that intend to evaluate both radiation dose rates and potential DNA damages to aquatic microorganisms and extend to other radioisotopes. The manuscript is well-written.

Response to Point 0 – We would like to thank the reviewer for their interest and the very encouraging review. 

Point 1, Lines 66 – 69 - “An exceptional abundance of deformations in the most radioactive springs in Auvergne has been recently revealed, initiating studies of the effects of natural radioactivity on benthic diatom communities in 16 mineral springs of the

area.”

Out of 16 mineral springs, 5 were chosen. What is the reason for choosing these 5 springs? Is it according to the degree of activity concentrations of 222Rn in water?

Response to Point 1 – We would like to thank the reviewer for their high interest in our research. These 5 locations were among the first to be identified and triggered our interest during the initial surveys in 2017. The research has been ongoing ever since, with valuable results presented in 2019 by Milan et al. which have led to more extensive studies in the region as presented by Baker et al. in 2022.

Point 2, Table 2 - Why the sampling date for Chateldon mineral spring is March 2017, not a specific date as those for other mineral spring? Is it because that the measured mass activities of radionuclides for Chateldon mineral spring is an average of multiple measurements in March?

Response to Point 2 – We would like to apologize for omitting the specific date of the sediment sampling in March 2017 at Chateldon. The sampling was performed on 01/03/2017 and it does not concern multiple measurements. Table 2 is now modified accordingly.

.

.

---

## [Decision Letter · Decision Letter 2]

22 Aug 2023

PONE-D-22-32535R2Simulation of the radiation exposure of microorganisms living in naturally radioactive mineral springs using GATE and Geant4-DNA Monte Carlo simulation toolsPLOS ONE

Dear Dr. Kolovi,

Thank you for submitting your manuscript to PLOS ONE. After careful consideration, we feel that it has merit but does not fully meet PLOS ONE’s publication criteria as it currently stands. Therefore, we invite you to submit a revised version of the manuscript that addresses the points raised during the review process.

We look forward to receiving your revised manuscript.

Kind regards,

Mohamad Syazwan Mohd Sanusi

Academic Editor

PLOS ONE

Journal Requirements:

Additional Editor Comments:

Dear author, thanks for the detail and complete revisions, the reviewers have agree to accept the revised version of your manuscript. However, before we proceed for final decision, I would like to invite the author for minor correction. Please find the attached file in this invitation:

Academic editor comments:

Title – Too concise. Double usage of “simulation” and it is too general. I believe titling an article would help the readers for their research interest and helping the keyword searching, plus increase the citation. The scope of work is in the dosimetry field. I would like to suggest the author to change the title to more technical one:

“Dosimetric technique for exposed microorganism to naturally radioactive mineral springs using GATE and Geant4-DNA Monte Carlo simulation”.

Or;

“Assessing Radiation Dosimetry for Microorganisms in Naturally Radioactive Mineral Springs using GATE and Geant4-DNA Monte Carlo Simulations”.

Abstract – The important part to give reader insights and highlighted finding of you work. Please include:

-radioactivity levels of Ra-226 & Rn-222 of the spring.

- a bit of technique used in this work (2-3 lines), what tools? Radiation source specs, any reference phantom, water sphere, simplified spherical geometry? Radiation transport physics treatment? Geant4 electromagnetic physics list option 4?

-your results (simulated dose rates), DNA damage counts/indicator and the threshold levels (ICRP or IAEA)?

- Revise this into separable text and please specify your investigated biota/species (eukaryotic photosynthetic 57 unicellular microalgae).

“We focus on the impact of the dominant radioelements, 226Ra in the sediments and 222Rn dissolved in the water of the mineral springs, and on diatoms, microalgae displaying an exceptional abundance of teratological forms in the most radioactive springs studied in Auvergne”.

-The conclusion is quite brief, any implication on the investigated species in future based on ICRP Publication 108 or related to Reference Animal in Publication 108.

Introduction, para 3 – please remove. “From the need of years-lasting………………….unique”. The texts given here doesn’t help reader gain insights of the purpose of the study. Please include the importance or the necessity of dose assessment for microbes/microorganisms. No solid technique for micro biotas dose assessment? For advancement in scientific knowledge of radioecology field?.

Line 52-53 – how high? Please include the activity levels.

Line 89 – chronic exposure to ionizing radiation.

Line 118 – please give specifically how many hours before gamma counting? To allow secular equilibrium? Please elaborate.

Line 178 – Justify why silicate shell need to model for diatom? My apology for mislook on any text that describe the anatomy of the studied diatom in section line 149 - 172, is there any elemental composition (SEM EDX) of these to justify the MC simulation diatoms phantoms?

Line 200-203 – The text cannot simply justify because of factor of low activity of U,Ra-228, Th in water spring for this study to choose Rn and Ra-226 only. There must a solid reason why this study chooses Rn and Ra over others. Line 49-53 already highlighted it.

Line 214 – an approximation is needed to extend the elaboration on the approach for radiation transport treatment. The question is; what is secondary electron energy range (control parameter) considered here for production cuts that stop tracking the particle further in Geant4 simulation. For smaller organisms, I believe the small energy contribution are important. Otherwise just state the approximation/simplification for physics treatment. Please elaborate the association of gammas? Because line 202 stated the consideration of alpha-particle only. Are they ancillary interaction of the alpha, please clarify.

Line 353 – revise accordingly as commented in line 200-203.

Line 360 – 362 - The statement of “external radiation exposure” simulation should be transferred in intro as well as in the abstract as it present the scope of this study.

Line 363 – redundant, please remove.

Conclusion – please conclude your work not summarizing it. You can directly point out on that average or range of absorbed dose received by the microorganism based on 3 simulation scenarios in 2-3 lines and conclude that the dose is compatible to threshold limit by ERICA (please use their original recommended levels by IAEA or ICRP), and conclude that the microorganism extremely exposed to DNA damages based on computed mutation rate or DSB or SSB.

Reviewers' comments:

Reviewer's Responses to Questions

**Comments to the Author**

1. If the authors have adequately addressed your comments raised in a previous round of review and you feel that this manuscript is now acceptable for publication, you may indicate that here to bypass the “Comments to the Author” section, enter your conflict of interest statement in the “Confidential to Editor” section, and submit your "Accept" recommendation.

Reviewer #3: All comments have been addressed

Reviewer #4: All comments have been addressed

2. Is the manuscript technically sound, and do the data support the conclusions?

Reviewer #3: Yes

Reviewer #4: Yes

3. Has the statistical analysis been performed appropriately and rigorously? 

Reviewer #3: Yes

Reviewer #4: Yes

4. Have the authors made all data underlying the findings in their manuscript fully available?

Reviewer #3: Yes

Reviewer #4: Yes

5. Is the manuscript presented in an intelligible fashion and written in standard English?

Reviewer #3: Yes

Reviewer #4: Yes

6. Review Comments to the Author

Reviewer #3: (No Response)

Reviewer #4: (No Response)

7. PLOS authors have the option to publish the peer review history of their article (what does this mean?). If published, this will include your full peer review and any attached files.

Reviewer #3: No

Reviewer #4: No

---

## [Author Response · Author response to Decision Letter 2]

24 Sep 2023

Dear editor,

We are grateful for having stood up to the expectations of the reviewers and we highly appreciate your interest in our work and the very productive comments. Please find below our corresponding responses, as well, as a proposed revised version of the manuscript.

We hope that this revision will improve the quality of the manuscript, meet the journal’s criteria, and merit publication.

Kind regards,

Kolovi Sofia on behalf of the authors

Academic editor comments:

Point 1, Title - Too concise. Double usage of “simulation” and it is too general. I believe titling an article would help the readers for their research interest and helping the keyword searching, plus increase the citation. The scope of work is in the dosimetry field. I would like to suggest the author to change the title to more technical one:

 “Dosimetric technique for exposed microorganism to naturally radioactive mineral springs using GATE and Geant4-DNA Monte Carlo simulation”.

Or;

“Assessing Radiation Dosimetry for Microorganisms in Naturally Radioactive Mineral Springs using GATE and Geant4-DNA Monte Carlo Simulations”.

Response to Point 1 – We fully embrace the need of modifying the title. The title was replaced by:

 “Assessing Radiation Dosimetry for Microorganisms in Naturally Radioactive Mineral Springs using GATE and Geant4-DNA Monte Carlo Simulations”

Point 2, Abstract - The important part to give reader insights and highlighted finding of you work. Please include: 

-radioactivity levels of Ra-226 & Rn-222 of the spring

- a bit of technique used in this work (2-3 lines), what tools? Radiation source specs, any reference phantom, water sphere, simplified spherical geometry? Radiation transport physics treatment? Geant4 electromagnetic physics list option 4?

-your results (simulated dose rates), DNA damage counts/indicator and the threshold levels (ICRP or IAEA)?

- Revise this into separable text and please specify your investigated biota/species (eukaryotic photosynthetic 57 unicellular microalgae).

“We focus on the impact of the dominant radioelements, 226Ra in the sediments and 222Rn dissolved in the water of the mineral springs, and on diatoms, microalgae displaying an exceptional abundance of teratological forms in the most radioactive springs studied in Auvergne”. 

-The conclusion is quite brief, any implication on the investigated species in future based on ICRP Publication 108 or related to Reference Animal in Publication 108.

Response to Point 2 – The abstract was revised according to the suggestions. We would like, nevertheless, to provide some clarifications:

- This work involves the collection of experimental data of 5 mineral springs. Nonetheless, the results focus on and are normalized to the reference values of 1000 Bq/L 222-Rn and 30 Bq/g 226-Ra, as stated in lines 141-144 of the manuscript, which do not explicit correspond to one specific mineral spring.

- Concerning the simulated microorganism, no correspondence to the Reference Animals and Plants of ICRP Publication 108 is feasible. There is still a lot of on-going work for diatoms, including their irradiation in controlled environments. 

The abstract was modified as follows:

“Mineral springs in Massif Central, France can be characterized by higher levels of

natural radioactivity in comparison to the background. The biota in these waters is

constantly under radiation exposure mainly from the α-emitters of the natural decay

chains, with 226Ra in sediments ranging from 21 Bq/g to 43 Bq/g and 222Rn activity

concentrations in water up to 4600 Bq/L. This study couples for the first time micro- and

nanodosimetric approaches to radioecology by combining GATE and Geant4-DNA to

assess the dose rates and DNA damages to microorganisms living in these naturally

radioactive ecosystems. It focuses on unicellular eukaryotic microalgae (diatoms) which

display an exceptional abundance of teratological forms in the most radioactive mineral

springs in Auvergne. Using spherical geometries for the microorganisms and based on

γ-spectrometric analyses, we evaluate the impact of the external exposure to 1000 Bq/L

222Rn dissolved in the water and 30 Bq/g 226Ra in the sediments. Our results show

that the external dose rates for diatoms are significant (9.7 μGy/h) and comparable to

the threshold (10 μGy/h) for the protection of the ecosystems suggested by the

literature. In a first attempt of simulating the radiation induced DNA damage on this

species, the rate of DNA Double Strand Breaks per day is estimated to 1.11E-04. Our

study confirms the significant mutational pressure from natural radioactivity to which

microbial biodiversity has been exposed since Earth origin in hydrothermal springs.”

Point 3, Introduction, Paragraph 3 - please remove. “From the need of years lasting ……….unique”. The texts given here doesn’t help reader gain insights of the purpose of the study. Please include the importance or the necessity of dose assessment for microbes/microorganisms. No solid technique for micro biotas dose assessment? For advancement in scientific knowledge of radioecology field?

Response to Point 3 – The indicated text has been removed. Additional text has been added in line 25 and line 40 in order to further highlight the importance and the necessity for micro biota dose assessments.

Line 25: …following the ICRP (International Commission on Radiological Protection) approach…

Line 40: Aiming to cover the needs of microscale radioecology,….

Point 4, Line 52-53 – how high? Please include the activity levels.

Response to Point 4 – Additional text was introduced as follows:

Lines 54 - 55: ...are found in high concentrations in the sediments (up to 31 Bq/g 226-Ra) and waters (up to 4600 Bq/L 222-Rn) of the mineral springs in Auvergne [35].

Point 5, Line 89 - chronic exposure to ionizing radiation.

Response to Point 5 - The text was modified accordingly in Line 91.

Point 6, Line 118 - please give specifically how many hours before gamma counting? To allow secular equilibrium? Please elaborate.

Response to Point 6 – γ- spectroscopy for 222-Rn dissolved in water was performed within the first 3 hours of sampling in order to allow for an integral measurement of the total radioactivity due to 214-Pb. This radon is not at secular equilibrium with 226-Ra in water. A second measurement can be done one month after sampling to measure 222-Rn sustained by 226-Ra in water that amounts to less than a few Bq/L.

The respective text was modified as follows:

Line 120: 222-Rn activity concentrations were measured within the first 3 hours after sampling …

Point 7, Line 178 - Justify why silicate shell need to model for diatom? My apology for mislook on any text that describe the anatomy of the studied diatom in section line 149 - 172, is there any elemental composition (SEM EDX) of these to justify the MC simulation diatoms phantoms?

Response to Point 7: Diatoms are the only known microorganisms with a rigid silicate frustule. Given the general morphology of diatoms described in lines 63 – 65, as well as, the highlighted importance of the frustule in lines 66 – 70 and lines 167 – 170, we assumed that the when the reader reaches line 180, he/she is already aware that diatoms are characterized by a silicate exoskeleton which is not encountered in any other microorganism. To improve the readability, we added the following text for clarification:

Line 181: …around the microorganism, representing their frustule (rigid exoskeleton).

Point 8, Lines 200 – 203: The text cannot simply justify because of factor of low activity of U, Ra-228, Th in water spring for this study to choose Rn and Ra-226 only. There must a solid reason why this study chooses Rn and Ra over others. Line 49-53 already highlighted it.

Response to Point 8: We thank the editor for this comment. Indeed, the intention of this text was to justify why the daughter nuclei of 226-Ra and 222-Rn were not simulated. Indeed, this study focuses on the dose rate to microorganisms coming from the radioelements that have been measured experimentally to have the highest activity in the sediments and waters of Auvergne springs. These radioelements also decay and their daughters contribute an additional radiation dose to the microorganisms. In a first step, this contribution was not computed using Monte-Carlo simulations because the chemical behaviour of these radionuclides and therefore their location in the vicinity of the diatoms is not known. As a result, the text was modified accordingly: 

Lines 202 – 212: “This study focuses on the dose rates to microorganisms coming from radioelements 198

that have been measured experimentally to have the highest activity concentrations in 199

the sediments and waters of Auvergne mineral springs. As a result, we simulated only 200

the α-particles emitted directly by 222Rn and 226Ra (see Table 4). These radioelements 201

also decay and their daughters, especially the α-emitters, contribute an additional 202

radiation dose to the microorganisms. In a first step, this contribution was not 203

computed using Monte-Carlo simulations because the chemical behaviour of these 204

radionuclides and therefore their location in the vicinity of the diatoms is not known."

Point 9, Line 214 - an approximation is needed to extend the elaboration on the approach for radiation transport treatment. The question is; what is secondary electron energy range (control parameter) considered here for production cuts that stop tracking the particle further in Geant4 simulation. For smaller organisms, I believe the small energy contribution are important. Otherwise just state the approximation/simplification for physics treatment. Please elaborate the association of gammas? Because line 202 stated the consideration of alpha-particle only. Are they ancillary interaction of the alpha, please clarify.

Response to Point 9: The application of productions cuts is used to avoid the tracking of very low energy secondary particles which is computational expensive. Various cuts were preliminary tested in order to evaluate the impact on the dose rates before concluding in the values presented in this manuscript. Thus, the following text was added to further clarify the use of the suggested production cuts:

Lines 223 – 225: The production cuts applied to secondary electrons and gammas, which are produced due to the interactions of the α-particles with the matter, were investigated in preliminary simulations, and chosen to be 2 orders of magnitude less than the size of the radius of the simulated volumes:…

Line 227: …,corresponding thus to the lowest cut-off energy available in GATE (250 eV).

Point 10, Line 353 - revise accordingly as commented in line 200-203.

Response to Point 10: We replaced the text “We chose….in five different mineral springs” in Lines 364-366 with the following:

Lines 366 - 372: We simulated separately the α-particles of 222-Rn in the water column and 226-Ra in the sediments of mineral springs in Auvergne, excluding their daughter nuclei in a first stage. Indeed, the contribution coming from the radioelements, especially the α-emitters in the decay chain of 226-Ra and 220-Rn depends on their location in the vicinity of the diatoms. Additional data are needed to understand their chemical behaviour and therefore their contribution to the dose rate to the microorganisms.

Point 11, Lines 360 – 362: The statement of “external radiation exposure” simulation should be transferred in intro as well as in the abstract as it presents the scope of this study.

Response to Point 11: The following text was added in order to highlight the simulation of external radiation exposure:

Abstract: … we evaluate the impact of the external exposure to …. Our results show that the external dose rates…. 

Introduction, Line 41: …for modelling the external radiation exposure and …

Point 12, Conclusion - please conclude your work not summarizing it. You can directly point out on that average or range of absorbed dose received by the microorganism based on 3 simulation scenarios in 2-3 lines and conclude that the dose is compatible to threshold limit by ERICA (please use their original recommended levels by IAEA or ICRP), and conclude that the microorganism extremely exposed to DNA damages based on computed mutation rate or DSB or SSB.

Response to Point 12: The conclusion was modified according to the suggestions. Part of the text was moved to the Discussion as follows:

Lines 455 – 458: With our work, we show that the computed mutation rate (4.7E-04 DSB/day) for the diatoms in the studied mineral springs is 4 orders of magnitude higher than for bacteria exposed solely to natural background radiation (2.8E-08 DSB/day) as calculated by Lampe et al. [78]

The Conclusion was modified as follows:

"In this study, we focused on the simulation of the external radiation exposure of 

microorganisms and diatoms living in naturally radioactive aquatic ecosystems. Three 

different environmental compositions were simulated corresponding to the ecosystems of 

mineral springs: dry sediments (0% porosity) containing only 30 Bq/g 226Ra, water 

(100% porosity) containing only 1000 Bq/L 222Rn and a benthic mixture of both (90% 

porosity) representing realistic conditions. In the benthic mixture, the diatom is 

exposed to 9.7 μGy/h due to 226Ra and 222Rn, a dose rate which is comparable with 

the threshold (10 μGy/h) for the protection of the ecosystems suggested by ERICA risk 

assessment tool. We evaluated that the frustule does not considerably protect diatoms 

from ionizing radiation. Based on our computed DSB, we show that the microorganisms 

are extremely exposed to DNA damages due to the chronic exposure to ionising 

radiation in the radioactive mineral springs. By demonstrating the coupling of 

experimental measurements with Monte Carlo simulations for two radionuclides, this 

work can be implemented in future radioecological studies wishing to estimate not only 

dose rates but also potential DNA damages on aquatic microorganisms and extend to 

other radioisotopes."

---

## [Editor Report · Decision Letter 3]

25 Sep 2023

Assessing radiation dosimetry for microorganisms in naturally radioactive mineral springs using GATE and Geant4-DNA Monte Carlo simulations

PONE-D-22-32535R3

Dear Dr. Sofia,

We’re pleased to inform you that your manuscript has been judged scientifically suitable for publication and will be formally accepted for publication once it meets all outstanding technical requirements.

Kind regards,

Mohamad Syazwan Mohd Sanusi

Academic Editor

PLOS ONE

Additional Editor Comments (optional):

Dear Dr. Sofia,

I wanted to express my appreciation for your diligent efforts in addressing the comments on the manuscript. Your thorough revision in response to my commnets has significantly improved the quality of the work. I am pleased to endorse the acceptance of revision 3 for publication.

Thank you for your dedication to this revision.

Best regards,
---

## [Editor Report · Acceptance letter]

2 Oct 2023

PONE-D-22-32535R3 

Assessing radiation dosimetry for microorganisms in naturally radioactive mineral springs using GATE and Geant4-DNA Monte Carlo simulations 

Dear Dr. Kolovi:

I'm pleased to inform you that your manuscript has been deemed suitable for publication in PLOS ONE. Congratulations! Your manuscript is now with our production department. 

Kind regards, 

on behalf of

Dr. Mohamad Syazwan Mohd Sanusi 

Academic Editor

PLOS ONE